# REALISTIC EVALUATION OF SEMI-SUPERVISED LEARNING ALGORITHMS IN OPEN ENVIRONMENTS

**Lin-Han Jia[1], Lan-Zhe Guo[1,3†], Zhi Zhou[1], Yu-Feng Li[1,2†]**
National Key Laboratory for Novel Software Technology[1]
School of Artificial Intelligence[2]
School of Intelligence Science and Technology[3]
Nanjing University, Nanjing 210023, China
{jialh,guolz,zhouz,liyf}@lamda.nju.edu.cn

## ABSTRACT

Semi-supervised learning (SSL) is a powerful paradigm for leveraging unlabeled data and has been proven to be successful across various tasks. Conventional SSL studies typically assume close environment scenarios where labeled and unlabeled examples are independently sampled from the same distribution. However, real-world tasks often involve open environment scenarios where the data distribution, label space, and feature space could differ between labeled and unlabeled data. This inconsistency introduces robustness challenges for SSL algorithms. In this paper, we first propose several robustness metrics for SSL based on the Robustness Analysis Curve (RAC), secondly, we establish a theoretical framework for studying the generalization performance and robustness of SSL algorithms in open environments, thirdly, we re-implement widely adopted SSL algorithms within a unified SSL toolkit and evaluate their performance on proposed open environment SSL benchmarks, including both image, text, and tabular datasets. By investigating the empirical and theoretical results, insightful discussions on enhancing the robustness of SSL algorithms in open environments are presented. The re-implementation and benchmark datasets are all publicly available. More details can be found at https://ygzwqzd.github.io/Robust-SSL-Benchmark.

## 1 INTRODUCTION

Semi-supervised learning (SSL) aims to leverage unlabeled data to improve learning performance when labels are limited or expensive to obtain (Chapelle et al., 2006). SSL algorithms have been repeatedly reported to achieve highly competitive performance to purely supervised learning and save a lot of labeling costs, by exploring the structure of unlabeled data.

All of the positive results, however, are based on the close environment assumption where labeled and unlabeled data are sampled from the same distribution independently. However, many practical applications involve open environments (Zhou, 2022) where the data distribution, feature space, and label space could be inconsistent between labeled and unlabeled data. SSL methods suffer severe robustness problems in open environments and could be even worse than a simple supervised learning model without exploiting more unlabeled data (Guo & Li, 2018; Oliver et al., 2018; Guo et al., 2020a; Li et al., 2021). Such phenomena undoubtedly go against the expectations of SSL and limit its effectiveness in more practical tasks.

The robustness of SSL in open environments has attracted great attention in recent years and various robust SSL algorithms have been proposed from different perspectives, such as inconsistent label space (Guo et al., 2020a; Chen et al., 2020; Yu et al., 2020; Saito et al., 2021; Guo & Li, 2022; Wei et al., 2022), inconsistent data distribution (Guo et al., 2020b; Zhou et al., 2021; Huang et al., 2021; Jia et al., 2023a). However, these algorithms primarily focus on robustness from a singular perspective and overlook the utilization of practical metrics for robustness analysis. Consequently, it remains challenging to ascertain the suitability of SSL algorithms in real-world open environments.

---

[†]Corresponding Authors: Lan-Zhe Guo and Yu-Feng Li.

In this paper, we first propose several metrics considering different aspects of performance in open environments to achieve a fair and comprehensive evaluation of SSL algorithms. Then, we establish a theoretical framework for studying the generalization performance and robustness of SSL algorithms, and the results show that the generalization error in SSL consists of five components: bias caused by the learner, variance caused by data sampling, and three types of inconsistencies caused by open environments. Finally, we re-implement widely adopted SSL algorithms within a unified SSL toolkit and evaluate their performance on proposed open environment SSL benchmarks, including both image, text, and tabular datasets. Some interesting findings include:

- Inconsistency between the feature and label space has a more detrimental impact compared to cases where there is inconsistency in data distribution.

- Traditional statistical SSL algorithms can often outperform deep SSL algorithms in terms of both performance and robustness when applied to tabular datasets. Thus, more advanced SSL algorithms on tabular datasets should be studied.

- Certain robust SSL algorithms currently proposed do not consistently exhibit enhanced robustness and may not surpass ordinary deep SSL algorithms in most scenarios. We argue that the robustness of SSL algorithms should be evaluated under more reasonable metrics.

- Inconsistency between labeled and unlabeled data does not invariably result in negative effects. On the contrary, leveraging inconsistent unlabeled examples may improve performance in some cases. Thus, it is important to study how to exploit helpful information from inconsistent unlabeled data.

## 2 ROBUST SSL IN OPEN ENVIRONMENTS

### 2.1 NOTATIONS

SSL algorithms leverage both labeled and unlabeled data for the learning process. In the close environments, all data generated from a consistent distribution $P(x, y), x \in \mathcal{X}, y \in \mathcal{Y}$ on consistent data space $\mathcal{X} \times \mathcal{Y} \subseteq \mathbb{R}^d \times \{1, \cdots, k\}$ where $d$ and $k$ respectively represent the number of features and classes. In SSL, we are given $n_l$ labeled examples $D_L = \{(x_i, y_i) | (x_i, y_i) \sim P(x, y)\}_{i=1}^{n_l}$ and $n_u$ unlabeled examples $D_U = \{x_i | x_i \sim P(x)\}_{i=1}^{n_u}$ where $P(x)$ is the marginal distribution of $P(x, y)$. The purpose is to learn a predictor with the smallest generalization error on $P(x, y)$.

In the open environments, we assume that all examples originate from a global data space $\mathcal{X}^* \times \mathcal{Y}^* \subseteq \mathbb{R}^{d^*} \times \{1, \ldots, k^*\}$ where $d^*$ and $k^*$ respectively represent the number of features and classes that appear throughout the entire learning process. There exists an invariant data distribution $P(y^*|x^*)$ for $x^* \in \mathcal{X}^*$ and $y^* \in \mathcal{Y}^*$. We denote the degree of inconsistency of data distribution, feature space, or label space between unlabeled and labeled data as $t \in [0, 1]$. A higher $t$ indicates a greater inconsistency. For any $t$, there is an inconsistent distribution denoted as $P_t(x^*, y^*)$. However, we can only obtain a projected distribution $P_t(x_t, y_t)$ in a subspace $\mathcal{X}_t \times \mathcal{Y}_t$, where $\mathcal{X}_t \subset \mathcal{X}^*$ and $\mathcal{Y}_t \subset \mathcal{Y}^*$.

We denote $\theta(t)$ as the function describing the ratio of inconsistent examples in the unlabeled dataset to $t$. For robust SSL with any $t$, we are given $n_l$ labeled examples from $P_0(x, y)$, $(1 - \theta(t)) \cdot n_u$ consistent unlabeled examples from $P_0(x)$ and $\theta(t) \cdot n_u$ inconsistent unlabeled examples from $P_t(x)$.

### 2.2 PERFORMANCE METRICS

To achieve a fair and comprehensive evaluation, we introduce a set of performance metrics tailored for robust SSL in open environments. These metrics begin by defining a function $Acc(t)$, which quantifies the change in model accuracy as a function of the inconsistency level $t$. This function is used to construct the Robustness Analysis Curve (RAC) that maps the inconsistency level $t$ to the corresponding accuracy $Acc(t)$. Unlike traditional SSL evaluations that focus solely on $Acc(0)$ or robust SSL evaluations that consider only a specific $Acc(t)$, our proposed metrics are derived from the RAC and provide a more comprehensive evaluation of SSL algorithms. These metrics include Area Under the RAC Curve (AUC) which captures the overall robustness of the SSL algorithm; Expected Accuracy (EA) which describes the average performance across all inconsistency levels; Worst-Case Accuracy (WA) which identifies the lowest accuracy level, representing the worst-case scenario; Expected Variation Magnitude (EVM) which captures the average magnitude of

Table 1: Performance Metrics for Robust Semi-Supervised Learning in Open Environments. $Acc(t)$ describe the change in model accuracy with the inconsistency extent $t$, $P_T(t)$ is the distribution for $t$, $Acc'(\cdot)$ indicate the first derivative.

| Area Under the Curve (AUC) | $AUC(Acc) = \int_0^1 Acc(t)dt$ |
|---|---|
| Expected Accuracy (EA) | $EA(P_T, Acc) = \langle P_T, Acc \rangle = \int_0^1 P_T(t)Acc(t)dt$ |
| Worst-Case Accuracy (WA) | $WA(Acc) = \min_{t \in [0,1]} Acc(t)$ |
| Expected Variation Magnitude (EVM) | $EVM(Acc) = \int_0^1 |Acc'(t)|dt$ |
| Variation Stability (VS) | $VS(Acc) = \int_0^1 [Acc'(t) - (\int_0^1 Acc'(t)dt)]^2 dt$ |
| Robust Correlation Coefficient (RCC) | $RCC(Acc) = \frac{\int_0^1 Acc(t) \cdot t dt - \int_0^1 Acc(t)dt}{\sqrt{\int_0^1 t^2 dt - 1} \cdot \sqrt{\int_0^1 Acc^2(t)dt - (\int_0^1 Acc(t)dt)^2}}$ |

performance variation; Variation Stability (VS) which quantifies the stability of the performance variation; Robust Correlation Coefficient (RCC) which captures the overall trend of performance variation. The detailed formulation of these metrics is presented in Table 1. By considering these diverse metrics, we can provide a comprehensive evaluation of the robustness of SSL algorithms, capturing different aspects of their performance. Moreover, these metrics are not limited to accuracy and can be extended to other performance measures by replacing the function $Acc(t)$.

## 2.3 DEFINITION OF ROBUSTNESS SSL IN OPEN-ENVIRONMENTS

Based on the proposed metrics, we propose a formal definition of the robust SSL in open environments, including the expected robustness and worst-case robustness.

**Definition 1.** *An SSL algorithm $\mathcal{A}$ returns a model $f_t \in \mathcal{F}$ using labeled data $D_L$ and unlabeled data $D_U^t$ for any inconsistency level $t$ where $\mathcal{F}$ is the hypothesis space of $\mathcal{A}$. Let $Acc(t)$ denote the accuracy of $f_t$ on the test data. When the inconsistency $t$ follows a distribution $P_T(t)$, if there exists $\sigma_E$ such that $|Acc(0) - EA(P_T, Acc)| \leq \sigma_E$, we say that A exhibits $\sigma_E$-expected algorithmic robustness. If there exists $\sigma_W$ such that $|Acc(0) - WA(Acc)| \leq \sigma_W$, we say that A exhibits $\sigma_W$-worst-case algorithmic robustness.*

In open environments, the SSL algorithm is employed to generate models with different inconsistency ratios $t$. If the algorithm can consistently deliver satisfactory performance across a range of $t$, we deem it to exhibit expected algorithmic robustness. If the algorithm can maintain acceptable performance levels in the worst case, we consider it to demonstrate worst-case algorithmic robustness.

## 3 THEORETICAL STUDIES ON ROBUST SSL

To analyze how to improve the robustness of algorithms in open environments, we establish a theoretical framework for studying the generalization performance and robustness of SSL algorithms in open environments.

Specifically, we first define the projection operations $\Pi_{\mathcal{X}}$ and $\Pi_{\mathcal{Y}}$ to project data distributions originating from different features and label spaces onto the same spaces with labeled data. Secondly, we formally define two types of inconsistencies: feature space inconsistency $Disc_F$ and label space inconsistency $Disc_L$, both of which represent additional generalization errors caused during the space projection process. Combined with the distribution inconsistency within the same data space $Disc_D$ defined in (Jia et al., 2023a), these constitute three types of inconsistencies in total. Finally, we analyze the SSL process in an open environment and ultimately conclude that the generalization error in SSL consists of five components: bias caused by the learner, variance caused by data sampling, and three types of inconsistencies caused by open environments.

**Theorem 3.1.** *For any target predictor $f \in \mathcal{F}$, pseudo-label predictor $h \in \mathcal{H}$, $0 \leq \delta_1 \leq 1$, $0 \leq \delta_2 \leq 1$ and $0 \leq \delta_3 \leq 1$, with the probability of at least $(1 - \delta_1)(1 - \delta_2)(1 - \delta_3)$:*

$$E(f, P_0(x,y)|h, w, map_{\mathcal{X}_t \to \mathcal{X}_0}, D_L, D_{U_t})$$
$$\leq \frac{n_l}{n_l + n_{u\,t}^w} \hat{E}(f, D_L) + \frac{n_{u\,t}^w}{n_l + n_{u\,t}^w} \hat{E}(f, \tilde{D}_{U_t}^w) + var(\mathcal{F}, n_l + n_{u\,t}^w, k_0, \delta_1)$$

$$+\frac{n_{u\,t}^w}{n_l+n_{u\,t}^w}(\theta^w(t)Disc_L(P_t^w(x^*),\mathcal{Y}_0)$$
$$+\theta^w(t)Disc_F(\Pi_{\mathcal{Y}_0}[P_t^w(x)],\Pi_{\mathcal{Y}_0}[P_t^w(x^*)],map_{\mathcal{X}_t\to\mathcal{X}_0},f)$$
$$+\theta^w(t)Disc_D(\Pi_{\mathcal{X}_0}[\Pi_{\mathcal{Y}_0}[P_t^w(x^*,y)]],P_0(x,y),f))$$
$$+\frac{n_{u\,t}^w}{n_l+n_{u\,t}^w}(\hat{E}(h,D_L)+var(\mathcal{H},n_l,k,\delta_2)+var(\mathcal{H},n_{u\,t}^w,k_0,\delta_3)$$
$$+\theta^w(t)Disc_L(P_t^w(x^*),\mathcal{Y}_0)+\theta^w(t)Disc_F(\Pi_{\mathcal{Y}_0}[P_t(x)],\Pi_{\mathcal{Y}_0}[P_t^w(x^*)],map_{\mathcal{X}_t\to\mathcal{X}_0},h)$$
$$+\theta^w(t)Disc_D(\Pi_{\mathcal{X}_0}[\Pi_{\mathcal{Y}_0}[P_t^w(x^*,y)]],P_0(x,y),h)) \tag{1}$$

*where $\hat{E}(f,\tilde{D}_{U_t}^w)$ is the weighted disagreement rate between the noisy pseudo-labels and the prediction results of $f$ on the weighted unlabeled dataset $\tilde{D}_{U_t}^w$.*

The conclusions drawn from theoretical analysis are as follows: inconsistencies in data distribution, feature space, and label space can all harm the generalization performance of the model. To alleviate the issue of data distribution inconsistency, it is primarily dependent on aligning the distributions based on the existing predictor, thereby optimizing the term $Disc_D$. To alleviate the issue of feature space inconsistency, it is primarily dependent on the feature mapping function, which requires the learning algorithm to accurately infer unobserved features based on the observed features, thereby optimizing the term $Disc_F$. To alleviate the issue of label space inconsistency, it primarily relies on sample selection and weighting functions, which require robust SSL algorithms to accurately detect and mitigate the negative impact of unfavorable examples, thereby optimizing the term $Disc_L$. The detailed theoretical analysis and proof are shown in appendix A.2.

## 4 EXPERIMENTS

### 4.1 EXPERIMENTAL SETUP

In our experiments, we evaluate both statistical SSL and deep SSL algorithms. For statistical SSL, we select 6 classical algorithms including the Semi-supervised Gaussian Mixture Model (SSGMM) (Shahshahani & Landgrebe, 1994) from the generative SSL algorithms, TSVM (Joachims et al., 1999) from the semi-supervised support vector machine algorithms, Label Propagation (Zhu & Ghahramani, 2003) and Label Spreading (Zhou et al., 2003) from the graph-based SSL algorithms, Tri-Training (Zhou & Li, 2005) from the disagreement-based SSL algorithms and Assemble (Bennett et al., 2002) from the ensemble-based SSL algorithms. For deep SSL, we select 10 representative algorithms: Pseudo Label (Lee, 2013), Pi-Model (Laine & Aila, 2017), Mean Teacher (Tarvainen & Valpola, 2017), ICT (Verma et al., 2022), VAT (Miyato et al., 2018), UDA (Xie et al., 2020), FixMatch (Sohn et al., 2020), FlexMatch (Zhang et al., 2021), FreeMatch (Wang et al., 2022b) and SoftMatch (Chen et al., 2023). We also considered 4 robust SSL algorithms: UASD (Chen et al., 2020), CAFA (Huang et al., 2021), MTCF (Yu et al., 2020), and Fix-A-Step (Huang et al., 2023).

We conduct experiments on various types of datasets, including 3 tabular datasets: iris, wine, letter; 3 image datasets: Image-CLEF (Caputo et al., 2014), CIFAR-10, and CIFAR-100; 3 text datasets: Amazon reviews (McAuley & Leskovec, 2013), IMDB movie reviews (Maas et al., 2011), and agnews (Zhang et al., 2015).

For all the experiments, we use mainstream supervised learning algorithms as baselines. For tabular data, we use XGBoost (Chen & Guestrin, 2016) as the benchmark for statistical learning algorithms and adopt FT-Transformer (Wang et al., 2022a) as the baseline and backbone for deep learning algorithms. For visual data, we use ResNet50 (He et al., 2016) as the baseline and backbone. For text data, we use the RoBERTa (Liu et al., 2019) model as the benchmark and backbone. All SSL algorithms are re-implemented based on the LAMDA-SSL toolkit (Jia et al., 2023b).

We plotted the RAC and performed statistical analysis on various evaluation metrics for different methods. For the plotting of the RAC curve, we sampled six $t$ values [0, 0.2, 0.4, 0.6, 0.8, 1] for all open environments. To ensure reliability, we conducted three experiments for each sampling point with seed values of $0\sim 2$. The average of these experiments was used to plot the curve. Linear interpolation was performed between adjacent sampling points. More detailed settings of the experiments are presented in appendix A.5.

Table 2: Evaluation of SSL algorithms using letter dataset under inconsistent data distributions

| Model | AUC | Acc(0) | WA | EVM | VS | RCC |
|---|---|---|---|---|---|---|
| XGBoost | 0.643 | 0.643 | 0.643 | - | - | - |
| TSVM | 0.624 | 0.650 | 0.607 | 0.012 | 0.012 | -0.733 |
| SSGMM | 0.276 | 0.334 | 0.245 | 0.022 | 0.029 | -0.740 |
| Label Propagation | 0.524 | 0.629 | 0.486 | 0.029 | 0.037 | -0.833 |
| Label Spreading | 0.588 | 0.653 | 0.563 | 0.020 | 0.025 | -0.780 |
| Tri-Training | 0.625 | 0.689 | 0.600 | 0.018 | 0.024 | -0.851 |
| Assemble | 0.644 | 0.646 | 0.641 | 0.003 | 0.003 | **0.037** |
| FT-Transformer | 0.660 | 0.660 | 0.660 | - | - | - |
| Pseudo Label | 0.658 | 0.667 | 0.652 | 0.003 | 0.002 | -0.977 |
| Pi-Model | 0.683 | 0.698 | 0.673 | 0.005 | 0.003 | -0.961 |
| Mean Teacher | 0.687 | 0.687 | 0.687 | **0.000** | **0.000** | - |
| VAT | **0.702** | 0.725 | 0.686 | 0.008 | 0.005 | -0.969 |
| ICT | 0.677 | 0.677 | 0.677 | **0.000** | **0.000** | - |
| UDA | 0.693 | 0.693 | 0.693 | **0.000** | **0.000** | - |
| FixMatch | 0.644 | **0.739** | 0.588 | 0.032 | 0.022 | -0.947 |
| FlexMatch | 0.649 | 0.687 | 0.618 | 0.014 | 0.010 | -0.973 |
| FreeMatch | 0.474 | 0.629 | 0.406 | 0.050 | 0.053 | -0.813 |
| SoftMatch | 0.584 | 0.654 | 0.564 | 0.020 | 0.032 | -0.664 |
| UASD | 0.701 | 0.702 | **0.700** | 0.001 | 0.002 | 0.008 |
| CAFA | 0.658 | 0.659 | 0.657 | 0.001 | 0.001 | -0.266 |
| MTCF | 0.365 | 0.612 | 0.270 | 0.081 | 0.102 | -0.702 |
| Fix-A-Step | 0.682 | **0.739** | 0.642 | 0.019 | 0.013 | -0.976 |

Table 3: Evaluation of deep SSL methods using ImageNet/Caltech dataset.

| Dataset | Model | AUC | Acc(0) | WA | EVM | VS | RCC |
|---|---|---|---|---|---|---|---|
| | Supervised | **0.909** | 0.909 | **0.909** | - | - | - |
| | Pseudo Label | 0.907 | 0.908 | 0.907 | 0.001 | 0.001 | -0.621 |
| | Pi-Model | **0.909** | 0.907 | 0.907 | 0.001 | 0.001 | 0.655 |
| | Mean Teacher | 0.903 | 0.904 | 0.900 | 0.003 | 0.003 | 0.169 |
| | VAT | 0.888 | 0.881 | 0.881 | 0.002 | 0.002 | **0.928** |
| | ICT | 0.907 | 0.909 | 0.903 | 0.003 | 0.004 | -0.359 |
| | UDA | 0.896 | 0.904 | 0.891 | 0.006 | 0.007 | -0.512 |
| ImageNet/Caltech | FixMatch | 0.902 | 0.905 | 0.887 | 0.005 | 0.007 | -0.726 |
| | FlexMatch | 0.906 | **0.921** | 0.893 | 0.008 | 0.010 | -0.861 |
| | FreeMatch | 0.864 | 0.916 | 0.832 | 0.031 | 0.028 | -0.786 |
| | SoftMatch | 0.904 | 0.908 | 0.891 | 0.007 | 0.007 | -0.805 |
| | UASD | 0.897 | 0.897 | 0.897 | **0.000** | **0.000** | - |
| | CAFA | 0.893 | 0.892 | 0.889 | 0.002 | 0.002 | 0.820 |
| | MTCF | 0.880 | 0.904 | 0.855 | 0.016 | 0.015 | -0.841 |
| | Fix-A-Step | 0.869 | 0.876 | 0.856 | 0.007 | 0.011 | -0.347 |

Table 4: Evaluation on IMDB/Amazon dataset with 100 labels under inconsistent data distributions

| Dataset | Model | AUC | Acc(0) | WA | EVM | VS | RCC |
|---|---|---|---|---|---|---|---|
| | Supervised | 0.571 | 0.571 | **0.571** | - | - | - |
| | Pseudo Label | **0.634** | 0.545 | 0.545 | 0.084 | 0.092 | 0.296 |
| | Pi-Model | 0.597 | **0.615** | 0.535 | 0.051 | 0.056 | 0.504 |
| | Mean Teacher | 0.601 | 0.570 | 0.538 | 0.096 | 0.101 | -0.012 |
| IMDB/Amazon | UDA | 0.599 | 0.523 | 0.523 | 0.071 | 0.080 | 0.484 |
| | FixMatch | 0.530 | 0.540 | 0.500 | 0.027 | 0.031 | -0.604 |
| | FlexMatch | 0.545 | 0.502 | 0.502 | 0.064 | 0.071 | 0.169 |
| | FreeMatch | 0.537 | 0.591 | 0.502 | 0.036 | 0.035 | -0.615 |
| | SoftMatch | 0.532 | 0.553 | 0.513 | **0.019** | **0.020** | -0.347 |
| | UASD | 0.593 | 0.541 | 0.541 | 0.046 | 0.055 | **0.580** |

Table 5: Evaluation of SSL algorithms using letter dataset under inconsistent feature space

| Model | AUC | Acc(0) | WA | EVM | VS | RCC |
|---|---|---|---|---|---|---|
| XGBoost | 0.694 | 0.694 | **0.694** | - | - | - |
| TSVM | 0.683 | **0.721** | 0.635 | 0.017 | 0.005 | -0.991 |
| SSGMM | 0.412 | 0.503 | 0.309 | 0.039 | 0.008 | -0.996 |
| Label Propagation | 0.589 | 0.642 | 0.542 | 0.020 | 0.012 | -0.978 |
| Label Spreading | 0.668 | 0.695 | 0.598 | 0.019 | 0.022 | -0.857 |
| Tri-Training | **0.696** | 0.716 | 0.664 | 0.010 | 0.008 | -0.945 |
| Assemble | 0.675 | 0.675 | 0.671 | 0.004 | 0.005 | **0.341** |
| FT-Transformer | 0.490 | 0.490 | 0.490 | - | - | - |
| Pseudo Label | 0.534 | 0.538 | 0.532 | 0.002 | 0.002 | -0.401 |
| Pi-Model | 0.541 | 0.552 | 0.537 | 0.003 | 0.004 | -0.761 |
| Mean Teacher | 0.517 | 0.517 | 0.517 | **0.000** | **0.000** | - |
| VAT | 0.541 | 0.561 | 0.535 | 0.007 | 0.008 | -0.720 |
| ICT | 0.540 | 0.540 | 0.540 | **0.000** | **0.000** | - |
| UDA | 0.537 | 0.537 | 0.537 | **0.000** | **0.000** | - |
| FixMatch | 0.499 | 0.548 | 0.470 | 0.022 | 0.035 | -0.237 |
| FlexMatch | 0.435 | 0.470 | 0.406 | 0.020 | 0.029 | -0.015 |
| FreeMatch | 0.447 | 0.409 | 0.409 | 0.014 | 0.008 | 0.977 |
| SoftMatch | 0.501 | 0.536 | 0.475 | 0.020 | 0.022 | -0.231 |
| UASD | 0.552 | 0.553 | 0.549 | 0.001 | 0.002 | -0.530 |
| CAFA | 0.511 | 0.511 | 0.510 | 0.001 | 0.001 | -0.358 |
| MTCF | 0.415 | 0.278 | 0.278 | 0.042 | 0.034 | 0.932 |
| Fix-A-Step | 0.511 | 0.561 | 0.490 | 0.020 | 0.032 | -0.311 |

## 4.2 SSL UNDER INCONSISTENT DATA DISTRIBUTIONS

We report the performance of SSL algorithms on letter, Image-CLEF, and IMDA/Amazon datasets in Table 2, Table 3, and Table 4, respectively. Results on more datasets are reported in appendix A.5. For tabular datasets, we calculate the centroids of each class and use the distance between examples and class centroids to filter examples, thus constructing an environment with inconsistent data distribution. For image and text datasets, we adopt the natural distribution shift to simulate the inconsistent distribution between labeled and unlabeled datasets.

## 4.3 SSL UNDER INCONSISTENT FEATURE SPACES

To simulate the inconsistent feature space, we randomly mask features for tabular data and each masked portion is filled with the mean value of the labeled data. For image datasets, we adopt the CIFAR-10 and CIFAR100 datasets and convert the images to grayscale, resulting in the loss of two color channels. For text data, we adopt the agnews (Zhang et al., 2015) dataset and employ text truncation. Truncated portions are filled with "$< pad >$" to simulate inconsistent feature spaces. The experimental results are reported in Table 5, Table 6, and Table 7.

## 4.4 SSL UNDER INCONSISTENT LABEL SPACES

The inconsistent label space between labeled and unlabeled data is the most widely studied problem in robust SSL. Following previous studies (Guo et al., 2020a; Oliver et al., 2018), we construct inconsistent labeled space by randomly selecting some classes and discarding the labeled data belonging to these classes for both tabular, image, and text datasets. The experimental results are reported in Table 8, Table 9, and Table 10.

## 4.5 EXPERIMENTAL RESULTS ANALYSIS

Based on the experimental results, we further conduct a comprehensive analysis from the perspectives of environments, algorithms, and performance metrics.

Table 6: Evaluation on CIFAR10 dataset under inconsistent feature spaces

| Dataset | Method | AUC | Acc(0) | WA | EVM | VS | RCC |
|---|---|---|---|---|---|---|---|
| CIFAR10 | Supervised | 0.473 | 0.473 | 0.473 | - | - | - |
| | Pseudo Label | 0.519 | 0.524 | **0.515** | **0.002** | **0.003** | -0.874 |
| | Pi-Model | 0.500 | 0.511 | 0.485 | 0.007 | 0.007 | -0.882 |
| | Mean Teacher | 0.470 | 0.486 | 0.457 | 0.006 | 0.005 | -0.962 |
| | VAT | 0.501 | 0.550 | 0.466 | 0.020 | 0.018 | -0.880 |
| | ICT | 0.468 | 0.476 | 0.456 | 0.005 | 0.005 | -0.929 |
| | UDA | 0.498 | 0.505 | 0.438 | 0.019 | 0.025 | -0.707 |
| | FixMatch | 0.517 | 0.551 | 0.430 | 0.037 | 0.042 | -0.661 |
| | FlexMatch | 0.552 | 0.607 | 0.431 | 0.041 | 0.039 | -0.921 |
| | FreeMatch | 0.555 | 0.645 | 0.423 | 0.045 | 0.029 | -0.962 |
| | SoftMatch | **0.559** | **0.661** | 0.453 | 0.042 | 0.009 | -0.998 |
| | UASD | 0.481 | 0.486 | 0.479 | 0.003 | **0.003** | -0.625 |
| | CAFA | 0.484 | 0.502 | 0.469 | 0.007 | **0.003** | -0.988 |
| | MTCF | 0.496 | 0.625 | 0.316 | 0.107 | 0.130 | **-0.604** |
| | Fix-A-Step | 0.516 | 0.551 | 0.424 | 0.025 | 0.032 | -0.832 |

Table 7: Evaluation on Agnews under inconsistent feature spaces

| Dataset | Method | AUC | Acc(0) | WA | EVM | VS | RCC |
|---|---|---|---|---|---|---|---|
| Agnews | Supervised | 0.844 | 0.844 | 0.844 | - | - | - |
| | Pseudo Label | 0.849 | 0.847 | 0.844 | 0.005 | 0.006 | **0.480** |
| | Pi-Model | 0.865 | 0.870 | 0.859 | **0.003** | **0.003** | -0.874 |
| | Mean Teacher | 0.851 | 0.856 | 0.841 | 0.004 | 0.004 | -0.890 |
| | UDA | 0.844 | 0.862 | 0.802 | 0.022 | 0.029 | -0.686 |
| | FixMatch | 0.870 | **0.880** | 0.858 | 0.005 | 0.005 | -0.944 |
| | FlexMatch | 0.848 | 0.877 | 0.810 | 0.021 | 0.021 | -0.829 |
| | FreeMatch | **0.876** | 0.872 | **0.868** | 0.008 | 0.009 | -0.131 |
| | SoftMatch | 0.875 | **0.880** | 0.865 | 0.005 | 0.005 | -0.815 |
| | UASD | 0.849 | 0.854 | 0.837 | 0.010 | 0.012 | -0.007 |

Table 8: Evaluation of SSL algorithms using letter dataset under inconsistent label space

| Model | AUC | Acc(0) | WA | EVM | VS | RCC |
|---|---|---|---|---|---|---|
| XGBoost | 0.694 | 0.694 | **0.694** | - | - | - |
| TSVM | 0.683 | **0.721** | 0.635 | 0.017 | **0.005** | -0.991 |
| SSGMM | 0.412 | 0.503 | 0.309 | 0.039 | 0.008 | -0.996 |
| Label Propagation | 0.589 | 0.642 | 0.542 | 0.02 | 0.012 | -0.978 |
| Label Spreading | 0.668 | 0.695 | 0.598 | 0.019 | 0.022 | -0.857 |
| Tri-Training | **0.696** | 0.716 | 0.664 | 0.010 | 0.008 | -0.945 |
| Assemble | 0.675 | 0.675 | 0.671 | 0.004 | 0.005 | 0.341 |
| FT-Transformer | 0.628 | 0.628 | 0.628 | - | - | - |
| Pseudo Label | 0.628 | 0.634 | 0.620 | 0.003 | 0.002 | -0.970 |
| Pi-Model | 0.649 | 0.653 | 0.639 | 0.005 | 0.007 | -0.673 |
| Mean Teacher | 0.635 | 0.635 | 0.635 | **0.000** | **0.000** | - |
| VAT | 0.640 | 0.656 | 0.622 | 0.007 | 0.004 | -0.984 |
| ICT | 0.607 | 0.607 | 0.607 | **0.000** | **0.000** | - |
| UDA | 0.606 | 0.605 | 0.605 | 0.001 | 0.001 | **0.657** |
| FixMatch | 0.602 | 0.663 | 0.556 | 0.021 | 0.012 | -0.983 |
| FlexMatch | 0.644 | 0.662 | 0.621 | 0.008 | 0.006 | -0.975 |
| FreeMatch | 0.528 | 0.634 | 0.447 | 0.042 | 0.041 | -0.937 |
| SoftMatch | 0.638 | 0.657 | 0.613 | 0.009 | 0.008 | -0.946 |
| UASD | 0.638 | 0.640 | 0.628 | 0.005 | 0.007 | 0.138 |
| CAFA | 0.620 | 0.622 | 0.615 | 0.002 | 0.002 | -0.918 |
| MTCF | 0.547 | 0.668 | 0.417 | 0.050 | 0.027 | -0.984 |
| Fix-A-Step | 0.648 | 0.668 | 0.614 | 0.013 | 0.008 | -0.966 |

Table 9: Evaluation on CIFAR10 under inconsistent label spaces

| Dataset | Method | AUC | Acc(0) | WA | EVM | VS | RCC |
|---------|--------|-----|--------|-----|-----|-----|-----|
| | Supervised | 0.643 | 0.643 | 0.643 | - | - | - |
| | Pseudo Label | 0.692 | 0.708 | 0.676 | 0.006 | 0.004 | -0.973 |
| | Pi-Model | 0.672 | 0.703 | 0.654 | 0.01 | 0.009 | -0.937 |
| | Mean Teacher | 0.639 | 0.647 | 0.634 | 0.003 | 0.005 | -0.333 |
| | VAT | 0.697 | 0.734 | 0.661 | 0.015 | 0.011 | -0.974 |
| | ICT | 0.643 | 0.647 | 0.642 | **0.002** | **0.002** | -0.819 |
| | UDA | 0.676 | 0.73 | 0.594 | 0.027 | 0.015 | -0.963 |
| CIFAR10 | FixMatch | 0.608 | 0.705 | 0.479 | 0.047 | 0.036 | -0.933 |
| | FlexMatch | 0.731 | 0.806 | 0.614 | 0.038 | 0.02 | -0.965 |
| | FreeMatch | 0.733 | **0.815** | 0.640 | 0.035 | 0.012 | -0.994 |
| | SoftMatch | 0.723 | 0.806 | 0.601 | 0.041 | 0.021 | -0.968 |
| | UASD | 0.644 | 0.641 | 0.641 | **0.002** | **0.002** | **0.404** |
| | CAFA | 0.675 | 0.674 | 0.672 | 0.005 | 0.006 | 0.093 |
| | MTCF | **0.747** | 0.798 | **0.681** | 0.024 | 0.008 | -0.989 |
| | Fix-A-Step | 0.681 | 0.757 | 0.517 | 0.048 | 0.048 | -0.908 |

Table 10: Evaluation on Agnews under inconsistent label spaces

| Dataset | Method | AUC | Acc(0) | WA | EVM | VS | RCC |
|---------|--------|-----|--------|-----|-----|-----|-----|
| | Supervised | 0.961 | 0.961 | **0.961** | - | - | - |
| | Pseudo Label | 0.960 | 0.956 | 0.956 | 0.007 | **0.006** | **0.307** |
| | Pi-Model | 0.962 | 0.968 | 0.950 | 0.006 | 0.006 | -0.785 |
| | Mean Teacher | **0.965** | 0.964 | **0.961** | **0.004** | **0.004** | 0.261 |
| Agnews | UDA | 0.956 | 0.965 | 0.938 | 0.010 | 0.009 | -0.816 |
| | FixMatch | 0.957 | **0.974** | 0.927 | 0.012 | 0.009 | -0.902 |
| | FlexMatch | 0.937 | 0.973 | 0.889 | 0.011 | 0.017 | -0.975 |
| | FreeMatch | 0.936 | 0.972 | 0.811 | 0.036 | 0.056 | -0.752 |
| | SoftMatch | 0.961 | **0.974** | 0.939 | 0.012 | 0.012 | -0.862 |
| | UASD | 0.954 | 0.948 | 0.944 | 0.013 | 0.014 | 0.112 |

**Environments**. We calculate the average expected robustness (under the uniform distribution of $P_T$) and the average worst-case robustness of SSL algorithms under different inconsistency settings. The results are reported in Table 11. According to the definition, lower values of $\sigma_E$ and $\sigma_W$ imply stronger robustness. The results show that the robustness of SSL algorithms is much lower in cases where there is inconsistency between the feature space and the label space, compared to cases when there is inconsistency in data distribution. Actually, inconsistencies between the feature and label spaces can both be considered as a greater degree of inconsistency in data distribution. The former can be viewed as a distribution shift where all missing features are assumed to take default values, while the latter can be seen as a distribution shift where the probability of all missing classes for examples is 0. These tell us that more attention needs to be paid to feature and label inconsistency between labeled and unlabeled data.

**Algorithms**. We compare the robustness of different algorithms in various environments and report the results in Table 12. We found that SSGMM (Shahshahani & Landgrebe, 1994) shows the poorest robustness among all algorithms, the main reason is it relies on the assumption of data distribution. For other statistical SSL algorithms, Assemble (Bennett et al., 2002) demonstrates the best performance and remarkable robustness, showcasing the advantage of using ensemble learning. For deep SSL algorithms, we find that the reported SOTA methods such as FixMatch, FlexMatch, SoftMatch, and FreeMatch, suffer severe robustness problems. One possible reason is that these methods adopt a threshold to select pseudo-labels for unlabeled data, which might overly centralize the distribution of unlabeled data. In comparison, UDA (Xie et al., 2020) sets thresholds for both labeled and unlabeled data, mitigating the inconsistency induced by sample selection to a large extent and significantly improving the robustness over FixMatch. For robust deep SSL algorithms, we find that UASD and CAFA achieve good robustness, but for other methods, they achieve lower robustness compared with

Table 11: Average Robustness of SSL Algorithms in Different Environments

| Environments | Excepted Robustness ($\bar{\sigma}_E$) | Worst-case Robustness ($\bar{\sigma}_W$) |
|---|---|---|
| Inconsistent Data Distributions | 0.015 | 0.028 |
| Inconsistent Feature Spaces | 0.019 | 0.039 |
| Inconsistent Label Spaces | 0.020 | 0.044 |

Table 12: Average Robustness of SSL Algorithms

| Algorithms | Excepted Robustness ($\bar{\sigma}_E$) | Worst-case Robustness ($\bar{\sigma}_W$) |
|---|---|---|
| SSGMM | 0.062 | 0.120 |
| TSVM | 0.017 | 0.040 |
| Label Propagation | 0.030 | 0.053 |
| Label Spreading | 0.021 | 0.045 |
| Tri-Training | 0.024 | 0.041 |
| Assemble | 0.009 | 0.017 |
| Pseudo Label | 0.008 | 0.014 |
| Pi-Model | 0.012 | 0.021 |
| Mean Teacher | 0.014 | 0.027 |
| VAT | 0.034 | 0.065 |
| ICT | 0.009 | 0.022 |
| UDA | 0.020 | 0.066 |
| FixMatch | 0.065 | 0.164 |
| FlexMatch | 0.055 | 0.143 |
| FreeMatch | 0.066 | 0.157 |
| SoftMatch | 0.067 | 0.154 |
| UASD | **0.003** | **0.002** |
| CAFA | 0.010 | 0.022 |
| MTCF | 0.077 | 0.118 |
| Fix-A-Step | 0.062 | 0.197 |

ordinary SSL algorithms. Therefore, when designing a robust SSL algorithm, we need to consider more comprehensive environments and evaluation metrics.

**Performance Metrics**. We also analyze the performance of different SSL algorithms under different metrics. First, we find that Acc(0) is not consistent with other metrics, SSL algorithms that have a high Acc(0) may perform even worse than supervised learning under the proposed robustness metrics. Second, we find that the EVM and VS metrics exhibit a high level of consistency, despite their different definitions. This indicates that for a robust SSL algorithm, its performance is less sensitive to changes in inconsistency level, and the direction of performance change is more stable and predictable. On the other hand, for a non-robust SSL algorithm, not only does it exhibit larger variations in performance, but the performance changes are also more unstable, showing greater randomness. Using such an algorithm in an open environment is extremely unsafe, as we cannot estimate its worst-case performance. Moreover, we find that the non-identically distributed unlabeled data is not always harmful, in some cases, exploiting more unlabeled data from inconsistent distributions may improve the performance. This inspires us to study SSL algorithms that fully exploit helpful information from inconsistent unlabeled data.

## 5 CONCLUSION

The research on robust SSL is an essential step toward the practical application of SSL. This paper provides a reshaped perspective on problem definition, performance metrics, theoretical frameworks, and evaluation of benchmark datasets. Our results provide evidence that SSL is still not robust in open environments, especially when the feature and labeled space are inconsistent between labeled and unlabeled data. These problems are often overlooked in previous studies and more efforts need to be devoted. The subsequent details about this work will be continuously supplemented and improved. We hope that our work can help push the successes of SSL towards the real world.

ACKNOWLEDGEMENTS

This research was supported by he National Science Foundation of China (62176118, 62306133).

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

# A APPENDIX

## A.1 LIMITATIONS

Although this paper has modeled and constructed complex open environments, it should be noted that the complexity of the real world may exceed the dataset we have constructed, and it may be difficult to evaluate, analyze, and explain with limited evaluation metrics and theoretical frameworks. For example, when multiple inconsistencies coexist and their degrees vary simultaneously, a high-dimensional vector $t$ rather than a one-dimensional variable $t$ is required to represent the combination of multiple inconsistencies. While the evaluation methods and metrics we employed can be readily extended to high-dimensional cases, this leads to an exponential increase in computational resource consumption, specifically on the order of $\Theta(s^{dim(t)})$, where $s$ represents the number of examples taken for each dimension. Finding ways to reduce the evaluation complexity in scenarios involving high-dimensional inconsistencies is an urgent and unresolved issue.

## A.2 THEORETICAL FRAMEWORK ON ROBUST SSL

Natarajan dimension(Natarajan, 1989) is an extension of Vapnik-Chervonen, dimension Vapnik & Chervonenkis (1971) in multi-classification problems. We denote $Ndim(\mathcal{H})$ the Natarajan dimension of a hypothesis space $\mathcal{H}$. To simplify the expression, we denote the variance term associated with the hypothesis space complexity in the generalization error with the number of examples $n$, the number of classes $k$, and the probability $\delta$:

$$var(\mathcal{H}, n, k, \delta) = \sqrt{\frac{16 Ndim(\mathcal{H}) \ln \sqrt{2n}k + 8 \ln \frac{2}{\delta}}{n}} \tag{2}$$

In an open environment, we assume that all data originates from a global space $\mathcal{X}^* \times \mathcal{Y}^* \subseteq \mathbb{R}^{d^*} \times \{0, \dots, k^* - 1\}$. There exists an invariant data distribution $P(y^*|x^*)$ for $x^* \in \mathcal{X}^*$ and $y^* \in \mathcal{Y}^*$. For any $t$, there is an inconsistent distribution denoted as $P_t(x^*, y^*)$. According to the total probability theorem, $P_t(x^*) = \sum_{y_i \in \mathcal{Y}^*}[P_t(y_i)P_t(x^*|y_i)]$ for all $x^* \in \mathcal{X}^*$.

However, we can only obtain a projected distribution $P_t(x_t, y_t)$ in a subspace $\mathcal{X}_t \times \mathcal{Y}_t$ from the global distribution $P_t(x^*, y^*)$ in the global space $\mathcal{X}^* \times \mathcal{Y}^*$, where $\mathcal{X}_t \subset \mathcal{X}^*$ and $\mathcal{Y}_t \subset \mathcal{Y}^*$. We denote $\bar{\mathcal{X}}_t = \mathcal{X}^*/\mathcal{X}_t$ as the unobserved features and $\bar{\mathcal{Y}}_t = \mathcal{Y}^*/\mathcal{Y}_t$ as the unobserved classes when the inconsistency rate is $t$. In this case, the observed inconsistent data follows the distribution $P_t(x) = \frac{\sum_{y_i \in \mathcal{Y}_t}(P_t(y_i)P_t(x^*|y_i))}{p_t(\bar{x}|x)}$, where $x \in \mathcal{X}_t \subseteq \mathbb{R}^{d_t}$ and $\bar{x} \in \bar{\mathcal{X}}_t \subseteq \mathbb{R}^{d^*-d_t}$, according to $\forall y_i \in \bar{\mathcal{Y}}_t, P_t(y_i) = 0$ and $P_t(x^*) = P_t(\bar{x}, x) = P_t(x)P_t(\bar{x}|x)$.

In SSL, labeled data are used to train a pseudo-label predictor $h \in \mathcal{H} : \mathcal{X}_0 \to \mathcal{Y}_0$ where $\mathcal{H}$ is the hypothesis space of pseudo label predictor to obtain unlabeled dataset with pseudo-labels, denoted as $\tilde{D}_U = \{(X_1^U, \tilde{y}_1^U), (X_2^U, \tilde{y}_2^U), \dots, (X_{n_u}^U, \tilde{y}_{n_u}^U)\}$. There is also a function $w : \mathcal{X}_0 \to \mathbb{R}$ used for sample weighting or selection. We denote the weighted unlabeled dataset without pseudo-labels as $D_U^w = w(D_U)$ and the weighted unlabeled dataset with noisy pseudo-labels as $\tilde{D}_U^w = w(\tilde{D}_U)$. We denote the sum of weights of all unlabeled examples as $n_u^w = \sum_{(x,y) \in D_U} w(x)$. We additionally denote the proportion of inconsistent examples in the unlabeled dataset after sample weighting as $\theta^w(t) = \frac{\sum_{i=(1-\theta(t))n_u+1}^{n_u} w(x_i)}{\sum_{i=1}^{n_u} w(x_i)}$. We define a weighted distribution as the inner product of a distribution function and a weighting function, such as $P^w(x) = w(x)P(x)$ and $P^w(x, y) = w(x)P(x, y)$.

SSL algorithms use both $D_L$ and $\tilde{D}_U^w$ for training a target predictor $f \in \mathcal{F}$ where $\mathcal{F}$ is the hypothesis space of the target predictor. Due to different feature spaces, feature mapping functions $map_{\mathcal{X}_t \to \mathcal{X}_0} : \mathcal{X}_t \to \mathcal{X}_0$ is also required to map the input into the domain of definition of the model.

For the distribution $P_t(x^*)$, we define its projection onto any target label space $\mathcal{Y}'$ as $\Pi_{\mathcal{Y}'}[P_t(x^*)] = \sum_{y_i \in \mathcal{Y}_t} \mathbb{I}(y_i \in \mathcal{Y}')P_t(y_i)P_t(x^*|y_i)$. The joint distribution after projecting onto the label space $\mathcal{Y}'$ is $\Pi_{\mathcal{Y}'}[P_t(x^*, y)] = (\sum_{y_i \in \mathcal{Y}_t} \mathbb{I}(y_i \in \mathcal{Y}')P_t(y_i)P_t(x^*|y_i))P(y|x^*)$.

For the distribution $P_t(x^*)$, we define its projection onto any feature space $\mathcal{X}'$ as $\Pi_{\mathcal{X}'}[P_t(x^*)] = P_t(map_{X^* \to \mathcal{X}'}(x^*)), x^* \sim P_t(x^*)$. The joint distribution after projecting onto the feature space $\mathcal{X}'$ is $\Pi_{\mathcal{X}'}[P_t(x^*, y)] = P_t(map_{X^* \to \mathcal{X}'}(x^*), y), (x^*, y) \sim P_t(x^*, y)$.

We define the discrepancy between the distribution $P_t(x^*)$ and the label space $\mathcal{Y}'$ as:

$$Disc_L(P_t(x^*), \mathcal{Y}') = 1 - \int_{x^* \sim P_t(x^*)} \Pi_{\mathcal{Y}'}[P_t(x^*)]dx^*. \tag{3}$$

For the data distribution $P_t(x^*)$ and the observed data $P_t(x)$, we define their discrepancy on an arbitrary feature space $\mathcal{X}'$ with respect to the feature mapping function $map_{\mathcal{X}_t \to \mathcal{X}'}$ and the model function f defined on the domain of $\mathcal{X}'$ as

$$Disc_F(P_t(x), P_t(x^*), map_{\mathcal{X}_t \to \mathcal{X}'}, f) = |P_{(x^*,y) \in P_t(x^*,y)}(f(map_{\mathcal{X}^* \to \mathcal{X}'}(x)) \neq y)$$
$$- P_{(x,y) \in P_t(x,y)}(f(map_{\mathcal{X}_t \to \mathcal{X}'}(x)) \neq y)|. \tag{4}$$

For two data distributions $P'(x,y)$ and $P''(x,y)$ defined on the same feature space and label space, their distributional difference to the model function f can be defined by the following discrepancy:

$$Disc_D(P'(x,y), P''(x,y), f) = |P_{(x,y) \sim P'(x,y)}(f(x) \neq y) - P_{(x,y) \sim P''(x,y)}(f(x) \neq y)| \tag{5}$$

As a result, we can obtain the error rate of pseudo-labeling in the weighted or filtered unlabeled dataset obtained through robust SSL.

**Theorem A.1.** *For any pseudo-label predictor $h \in \mathcal{H}$, $0 \leq \delta_1 \leq 1$ and $0 \leq \delta_2 \leq 1$, with the probability of at least $(1 - \delta_1)(1 - \delta_2)$:*

$$\hat{E}(h, w, map_{\mathcal{X}_t \to \mathcal{X}_0}, D_{U_t})$$
$$\leq \hat{E}(h, D_L) + var(\mathcal{H}, n_l, k_0, \delta_1) + var(\mathcal{H}, n_{u\,t}^w, k_0, \delta_2) + \theta^w(t)Disc_L(P_t^w(x^*), \mathcal{Y}_0)$$
$$+ \theta^w(t)Disc_F(\Pi_{\mathcal{Y}_0}[P_t^w(x)], \Pi_{\mathcal{Y}_0}[P_t^w(x^*)], map_{\mathcal{X}_t \to \mathcal{X}_0}, h)$$
$$+ \theta^w(t)Disc_D(\Pi_{\mathcal{X}_0}[\Pi_{\mathcal{Y}_0}[P_t^w(x^*, y)]], P_0(x, y), h) \tag{6}$$

*where $\hat{E}(h, D_L)$ is the empirical error of h on $D_L$ and $\hat{E}(h, w, map_{\mathcal{X}_t \to \mathcal{X}_0}, D_{U_t})$ is the empirical error of h on $D_U$ with ground truth labels.*

Based on the above label noise rate bound of the unlabeled dataset, we can estimate the generalization error bound of the robust SSL algorithm trained with the labeled dataset and this unlabeled dataset.

**Theorem A.2.** *For any target predictor $f \in \mathcal{F}$, pseudo-label predictor $h \in \mathcal{H}$, $0 \leq \delta_1 \leq 1$, $0 \leq \delta_2 \leq 1$ and $0 \leq \delta_3 \leq 1$, with the probability of at least $(1 - \delta_1)(1 - \delta_2)(1 - \delta_3)$:*

$$E(f, P_0(x, y)|h, w, map_{\mathcal{X}_t \to \mathcal{X}_0}, D_L, D_{U_t})$$
$$\leq \frac{n_l}{n_l + n_{u\,t}^w} \hat{E}(f, D_L) + \frac{n_{u\,t}^w}{n_l + n_{u\,t}^w} \hat{E}(f, \tilde{D}_{U_t}^w) + var(\mathcal{F}, n_l + n_{u\,t}^w, k_0, \delta_1)$$
$$+ \frac{n_{u\,t}^w}{n_l + n_{u\,t}^w}(\theta^w(t)Disc_L(P_t^w(x^*), \mathcal{Y}_0)$$
$$+ \theta^w(t)Disc_F(\Pi_{\mathcal{Y}_0}[P_t^w(x)], \Pi_{\mathcal{Y}_0}[P_t^w(x^*)], map_{\mathcal{X}_t \to \mathcal{X}_0}, f)$$
$$+ \theta^w(t)Disc_D(\Pi_{\mathcal{X}_0}[\Pi_{\mathcal{Y}_0}[P_t^w(x^*, y)]], P_0(x, y), f))$$
$$+ \frac{n_{u\,t}^w}{n_l + n_{u\,t}^w}(\hat{E}(h, D_L) + var(\mathcal{H}, n_l, k, \delta_2) + var(\mathcal{H}, n_{u\,t}^w, k_0, \delta_3)$$
$$+ \theta^w(t)Disc_L(P_t^w(x^*), \mathcal{Y}_0) + \theta^w(t)Disc_F(\Pi_{\mathcal{Y}_0}[P_t(x)], \Pi_{\mathcal{Y}_0}[P_t^w(x^*)], map_{\mathcal{X}_t \to \mathcal{X}_0}, h)$$
$$+ \theta^w(t)Disc_D(\Pi_{\mathcal{X}_0}[\Pi_{\mathcal{Y}_0}[P_t^w(x^*, y)]], P_0(x, y), h)) \tag{7}$$

*where $\hat{E}(f, \tilde{D}_{U_t}^w)$ is the weighted disagreement rate between the noisy pseudo-labels and the prediction results of f on the weighted unlabeled dataset $\tilde{D}_{U_t}^w$.*

## A.3 FORMULA DERIVATION AND THEORETICAL PROOF

### A.3.1 DERIVATION OF EVALUATION METRICS

Since the formulas for the metrics AUC, EA, WA, and EVM can be directly obtained through definitions, we primarily focus on deriving the formulas for the metrics VS and RCC.

1. VS: According to the definition, the metric VS is used to measure the stability of model performance changes, that is whether the derivative of $Acc(t)$, $Acc'(t)$, fluctuates significantly. We describe the magnitude of fluctuations using the variance, and thus, VS is defined as the variance of $Acc'(t)$. We denote the expectation of the variable x as $E(x)$ and the standard deviation of the variable x as $\sigma(X)$.

$$
\begin{aligned}
& VS(Acc) \\
&= \sigma^2(Acc') \\
&= \int_0^1 [Acc'(t) - E(Acc')]^2 dt \\
&= \int_0^1 [Acc'(t) - (\int_0^1 Acc'(t)dt)]^2 dt
\end{aligned}
\tag{8}
$$

2. RCC: According to the definition, the metric RCC is used to measure the correlation between model performance and the inconsistency factor $t$. The Pearson correlation coefficient effectively quantifies the correlation between two variables. We denote the covariance between the variables x and Y as $COV(x)$ and the Pearson correlation coefficient between the variables x and Y as $\rho(X, Y)$.

$$
\begin{aligned}
& \rho(X, Y) \\
&= \frac{COV(X, Y)}{\sigma(X)\sigma(Y)} \\
&= \frac{E(XY) - E(X)E(Y)}{\sqrt{E(X^2) - E^2(X)}\sqrt{E(Y^2) - E^2(Y)}}
\end{aligned}
\tag{9}
$$

We can directly apply the formula for the Pearson correlation coefficient.

$$
\begin{aligned}
& RCC(Acc) \\
&= \rho(Acc, t) \\
&= \frac{E(Acc \cdot t) - E(Acc)E(t)}{\sqrt{E(Acc^2) - E^2(Acc)}\sqrt{E(t^2) - E^2(t)}} \\
&= \frac{\int_0^1 Acc(t) \cdot t dt - \int_0^1 Acc(t)dt \int_0^1 t dt}{\sqrt{\int_0^1 Acc^2(t)dt - (\int_0^1 Acc(t)dt)^2} \cdot \sqrt{\int_0^1 t^2 dt - (\int_0^1 t dt)^2}} \\
&= \frac{\int_0^1 Acc(t) \cdot t dt - \int_0^1 Acc(t)dt}{\sqrt{\int_0^1 Acc^2(t)dt - (\int_0^1 Acc(t)dt)^2} \cdot \sqrt{\int_0^1 t^2 dt - 1}}
\end{aligned}
\tag{10}
$$

### A.3.2 Proof of Theoretical Results

Proof of Theorem 1

In the case of using only $n_l$ labeled examples for supervised learning, for any $h \in \mathcal{H}$ and $0 \leq \delta_1 \leq 1$, with the probability of at least $1 - \delta_1$:

$$
E(h, P_0(x, y)) \leq \hat{E}(h, D_L) + var(\mathcal{H}, n_l, k, \delta_1)
\tag{11}
$$

where $\hat{E}(h, D_L)$ is the empirical error of $h$ on the dataset $D_L$ and $E(f, P_0(x, y))$ is the generalization error of $f$ on the distribution of labeled data $P_0(x, y)$.

In SSL, when all examples are from the same distribution, for any $t$, dataset $D_{U_t}$ with $n_u$ examples are from the same distribution $P_t(x, y) = P_0(x, y)$. For any $h \in \mathcal{H}$ and $0 \leq \delta_2 \leq 1$, with the probability of at least $1 - \delta_2$:

$$
\begin{aligned}
\hat{E}(h, D_U^t) &\leq E(h, P_t(x, y)) + var(\mathcal{H}, n_u, k, \delta_2) \\
&= E(h, P_0(x, y)) + var(\mathcal{H}, n_u, k, \delta_2)
\end{aligned}
\tag{12}
$$

According to eqs. (11) and (12), for any pseudo-label predictor $h \in \mathcal{H}$, $0 \le \delta_1 \le 1$ and $0 \le \delta_2 \le 1$, with the probability of at least $(1 - \delta_1)(1 - \delta_2)$:

$$\hat{E}(h, D_{U_t}) \le \hat{E}(h, D_L) + var(\mathcal{H}, n_l, k, \delta_1) + var(\mathcal{H}, n_u, k, \delta_2) \tag{13}$$

When labeled data and unlabeled data are from different distributions, for any $h \in \mathcal{H}$:

$$
\begin{aligned}
&E(h, P_t(x, y)) \\
&\le E(h, P_0(x, y)) + |P_{x,y \sim P_0(x,y)}(h(x) \ne y) - P_{x,y \sim P_t(x,y)}(h(x) \ne y)| \\
&= E(h, P_0(x, y)) + Disc(h, P_0(x, y), P_t(x, y))
\end{aligned}
\tag{14}
$$

According to eqs. (11), (12) and (14), for any pseudo-label predictor $h \in \mathcal{H}$, $0 \le \delta_1 \le 1$ and $0 \le \delta_2 \le 1$, with the probability of at least $(1 - \delta_1)(1 - \delta_2)$:

$$
\begin{aligned}
&\hat{E}(h, P_t(x, y)) \\
&\le E(h, P_0(x, y)) + Disc_D(P_0(x, y), P_t(x, y), h) \\
&\le \hat{E}(h, D_L) + var(\mathcal{H}, n_l, k, \delta_1) + var(\mathcal{H}, n_u, k, \delta_2) + Disc_D(P_0(x, y), P_t(x, y), h)
\end{aligned}
\tag{15}
$$

Taking into account that in SSL, a weighting function $w$ is often used to either weigh or filter unlabeled examples, it's the weighted unlabeled data that truly plays a role in the learning process.

$$
\begin{aligned}
&\hat{E}(h, D_{U_t}, w) \\
&\le E(h, P_0(x, y)) + Disc_D(P_0(x, y), w(P_t(x, y)), h) \\
&\le \hat{E}(h, D_L) + var(\mathcal{H}, n_l, k, \delta_1) + var(\mathcal{H}, n_{u\,t}^w, k, \delta_2) + Disc_D(P_0(x, y), P_t^w(x, y), h)
\end{aligned}
\tag{16}
$$

Now considering that labeled and unlabeled data are not only from inconsistent data distributions but also inconsistent data spaces, we need an extra feature mapping function to complete the features and an extra weighting function to filter out examples from new classes. Both the mapping function and the weighting function aim to project unlabeled data to the same space as labeled data.

According to eqs. (3), (4) and (16), for any pseudo-label predictor $h \in \mathcal{H}$, $0 \le \delta_1 \le 1$ and $0 \le \delta_2 \le 1$, with the probability of at least $(1 - \delta_1)(1 - \delta_2)$:

$$
\begin{aligned}
&\hat{E}(h, w, map_{\mathcal{X}_t \to \mathcal{X}_0}, D_{U_t}) \\
&\le \hat{E}(h, D_L) + var(\mathcal{H}, n_l, k_0, \delta_1) + var(\mathcal{H}, n_{u\,t}^w, k_0, \delta_2) + \theta^w(t) Disc_L(P_t^w(x^*), \mathcal{Y}_0) \\
&+ \theta^w(t) Disc_F(\Pi_{\mathcal{Y}_0}[P_t^w(x)], \Pi_{\mathcal{Y}_0}[P_t^w(x^*)], map_{\mathcal{X}_t \to \mathcal{X}_0}, h) \\
&+ \theta^w(t) Disc_D(P_0(x, y), \Pi_{\mathcal{X}_0}[\Pi_{\mathcal{Y}_0}[P_t^w(x^*, y)]], h)
\end{aligned}
\tag{17}
$$

where $\hat{E}(h, D_L)$ is the empirical error of $h$ on $D_L$ and $\hat{E}(h, w, map_{\mathcal{X}_t \to \mathcal{X}_0}, D_{U_t})$ is the empirical error of $h$ on $D_{U_t}$ with ground truth labels.

Proof of Theorem 2

We denote the mixture of two distributions $\mathcal{D}_1$ and $\mathcal{D}_2$ with proportion $\alpha$ as:

$$Mix_\alpha(\mathcal{D}_1, \mathcal{D}_2) = \alpha \mathcal{D}_1 + (1 - \alpha)\mathcal{D}_2 \tag{18}$$

In SSL with inconsistent distributions, the target predictor is trained with both labeled dataset $D_L$ and weighted unlabeled dataset with noisy pseudo-labels $\tilde{D}_{U_t}$. $D_L$ and $\tilde{D}_{U_t}$ can be considered as a mixed dataset with $n_l + n_{u_t}$ examples from the mixed distribution $Mix_{\frac{n_l}{n_l + n_{u_t}}}(P_0(x, y), P_t(x, y))$ whose noisy rate is $\frac{n_{u_t}}{n_l + n_{u_t}} \hat{E}(h, D_{Ut})$.

So, for any target predictor $f \in \mathcal{F}$, pseudo-label predictor $h \in \mathcal{H}$, $0 \le \delta_3 \le 1$, with the probability of at least $1 - \delta_3$:

$$
\begin{aligned}
&E(f, Mix_{\frac{n_l}{n_l + n_{u_t}}}(P_0(x, y), P_t(x, y))|h, D_L, D_U) \\
&\le \frac{n_l}{n_l + n_{u_t}} \hat{E}(f, D_L) + \frac{n_{u_t}}{n_l + n_{u_t}} \hat{E}(f, \tilde{D}_U) + var(\mathcal{F}, n_l + n_{u_t\,t}, k, \delta_3) + \frac{n_{u_t}}{n_l + n_{u_t}} \hat{E}(h, D_U)
\end{aligned}
\tag{19}
$$

where $E(f, Mix_{\frac{n_l}{n_l+n_{u_t}}}(P_0(x,y), P_t(x,y))|h, D_L, D_U)$ is the generalization error of $f$ on the distribution $Mix_{\frac{n_l}{n_l+n_{u_t}}}(P_0(x,y), P_t(x,y)$ corresponding to pseudo-label predictor $h$.

When labeled data and unlabeled data are from different distributions, for any $f \in \mathcal{F}$:

$$
\begin{aligned}
&E(f, P_0(x,y)|h, D_L, D_{U_t}) \\
\leq &E(f, Mix_{\frac{n_l}{n_l+n_{u_t}}}(P_0(x,y), P_t(x,y))|h, D_L, D_{U_t}) \\
&+ |p_{x,y \sim P_0(x,y)}(h(x) \neq y) - p_{x,y \sim Mix_{\frac{n_l}{n_l+n_{u_t}}}(P_0(x,y), P_t(x,y))}(h(x) \neq y)| \\
= &E(f, Mix_{\frac{n_l}{n_l+n_{u_t}}}(P_0(x,y), P_t(x,y))|h, D_L, D_{U_t}) \\
&+ Disc_D(f, P_0(x,y), Mix_{\frac{n_l}{n_l+n_{u_t}}}(P_0(x,y), P_t(x,y)))
\end{aligned}
\tag{20}
$$

According to eqs. (16), (19) and (20), for any target predictor $f \in \mathcal{F}$, pseudo-label predictor $h \in \mathcal{H}$, $0 \leq \delta_1 \leq 1$, $0 \leq \delta_2 \leq 1$ and $0 \leq \delta_3 \leq 1$, with the probability of at least $(1-\delta_1)(1-\delta_2)(1-\delta_3)$:

$$
\begin{aligned}
&E(f, P_0(x,y)|h, D_L, D_{U_t}) \\
\leq &E(f, Mix_{\frac{n_l}{n_l+n_{u_t}}}(P_0(x,y), P_t(x,y))|h, D_L, D_{U_t}) \\
&+ Disc_D(f, P_0(x,y), Mix_{\frac{n_l}{n_l+n_{u_t}}}(P_0(x,y), P_t(x,y))) \\
\leq &\frac{n_l}{n_l+n_{u_t}}\hat{E}(f, D_L) + \frac{n_{u_t}}{n_l+n_{u_t}}\hat{E}(f, \tilde{D}_{U_t}) + var(\mathcal{F}, n_l+n_{u_t}, k, \delta_3) \\
&+ \frac{n_{u_t}}{n_l+n_{u_t}}\hat{E}(h, D_{U_t}) + Disc_D(f, P_0(x,y), Mix_{\frac{n_l}{n_l+n_{u_t}}}(P_0(x,y), P_t(x,y))) \\
\leq &\frac{n_l}{n_l+n_{u_t}}\hat{E}(f, D_L) + \frac{n_{u_t}}{n_l+n_{u_t}}\hat{E}(f, \tilde{D}_{U_t}) + var(\mathcal{F}, n_l+n_{u_t}, k, \delta_1) \\
&+ Disc_D(f, P_0(x,y), Mix_{\frac{n_l}{n_l+n_{u_t}}}(P_0(x,y), P_t(x,y))) \\
&+ \frac{n_{u_t}}{n_l+n_{u_t}}(\hat{E}(h, D_L) + var(\mathcal{H}, n_l, k, \delta_2) + var(\mathcal{H}, n_{u_t}, k, \delta_3) + Disc_D(h, P_0(x,y), P_t(x,y)))
\end{aligned}
\tag{21}
$$

where $\hat{E}(f, \tilde{D}_{U_t})$ is the weighted empirical inconsistency rate between the noisy pseudo-labels and the prediction results of $f$ on the unlabeled dataset $\tilde{D}_{U_t}$.

Taking into account inconsistent label spaces and weighting function $w$:

$$
\begin{aligned}
&E(f, P_0(x,y)|h, D_L, D_{U_t}, w) \\
\leq &\frac{n_l}{n_l+n_{u_t}^w}\hat{E}(f, D_L) + \frac{n_{u_t}^w}{n_l+n_{u_t}^w}\hat{E}(f, \tilde{D}_{U_t}^w) + var(\mathcal{F}, n_l+n_{u_t}^w, k, \delta_1) \\
&+ Disc_D(f, P_0(x,y), Mix_{\frac{n_l}{n_l+n_{u_t}^w}}(P_0(x,y), P_t^w(x,y))) \\
&+ \frac{n_{u_t}^w}{n_l+n_{u_t}^w}(\hat{E}(h, D_L) + var(\mathcal{H}, n_l, k, \delta_2) + var(\mathcal{H}, n_{u_t}^w, k, \delta_3) \\
&+ Disc_D(h, P_0(x,y), P_t^w(x,y)))
\end{aligned}
\tag{22}
$$

Taking into account inconsistent feature spaces and mapping function $map_{\mathcal{X}_t \to \mathcal{X}_0}$, the final error bound can be obtained.

According to eqs. (3), (4), (17), (21) and (22), for any target predictor $f \in \mathcal{F}$, pseudo-label predictor $h \in \mathcal{H}$, $0 \leq \delta_1 \leq 1$, $0 \leq \delta_2 \leq 1$ and $0 \leq \delta_3 \leq 1$, with the probability of at least $(1-\delta_1)(1-\delta_2)(1-\delta_3)$:

$E(f, P_0(x,y)|h, w, map_{\mathcal{X}_t \to \mathcal{X}_0}, D_L, D_U^t)$

$$\leq \frac{n_l}{n_l + n_{u_t}^w} \hat{E}(f, D_L) + \frac{n_{u_t}^w}{n_l + n_{u_t}^w} \hat{E}(f, \tilde{D}_{U_t}^w)) + var(\mathcal{F}, n_l + n_{u_t}^w, k_0, \delta_1)$$

$$+ \frac{n_{u_t}^w}{n_l + n_{u_t}^w} (\theta^w(t) Disc_L(P_t^w(x^*), \mathcal{Y}_0) + \theta^w(t) Disc_F(\Pi_{\mathcal{Y}_0}[P_t^w(x)], \Pi_{\mathcal{Y}_0}[P_t^w(x^*)], map_{\mathcal{X}_t \to \mathcal{X}_0}, f)$$

$$+ \theta^w(t) Disc_D(\Pi_{\mathcal{X}_0}[\Pi_{\mathcal{Y}_0}[P_t^w(x^*, y)]], P_0(x, y), f))$$

$$+ \frac{n_{u_t}^w}{n_l + n_{u_t}^w} (\hat{E}(h, D_L) + var(\mathcal{H}, n_l, k, \delta_2) + var(\mathcal{H}, n_{u_t}^w, k_0, \delta_3)$$

$$+ \theta^w(t) Disc_L(P_t^w(x^*), \mathcal{Y}_0) + \theta^w(t) Disc_F(\Pi_{\mathcal{Y}_0}[P_t(x)], \Pi_{\mathcal{Y}_0}[P_t^w(x^*)], map_{\mathcal{X}_t \to \mathcal{X}_0}, h)$$

$$+ \theta^w(t) Disc_D(\Pi_{\mathcal{X}_0}[\Pi_{\mathcal{Y}_0}[P_t^w(x^*, y)]], P_0(x, y), h)) \tag{23}$$

where $\hat{E}(f, \tilde{D}_{U_t}^w)$ is the weighted disagreement rate between the noisy pseudo-labels and the prediction results of $f$ on the weighted unlabeled dataset $\tilde{D}_{U_t}^w$.

## A.4 SSL Algorithms Evaluated in the benchmark

### A.4.1 Statistical SSL Algorithms

1. SSGMM (Shahshahani & Landgrebe, 1994) assumes that data is generated by a Gaussian mixture model, that is, the marginal distribution of examples can be expressed as the result of mixing several Gaussian distributions, and each distribution is given a weight.

2. TSVM (Joachims et al., 1999) infers labels of unlabeled examples and finds dividing hyperplanes that maximize the distance from support vectors. It continuously finds pairs of unlabeled heterogeneous examples and exchanges their labels until no more pairs can be found.

3. Label Propagation (Zhu & Ghahramani, 2003) uses examples as nodes, and the relationship between the examples as edges. The purpose of the Label Propagation algorithm is to propagate the labels from labeled data to unlabeled data through the graph.

4. Label Spreading (Zhou et al., 2003) penalizes misclassified labeled examples rather than banning misclassification completely which is different from Label Propagation fixing labels of labeled examples during the spreading process.

5. Tri-Training (Zhou & Li, 2005) is a representative disagreement-based SSL algorithm. It uses three learners with divergence and makes divergence by data sampling. The disagreement between learners is utilized for optimizing interactively.

6. Assemble (Bennett et al., 2002) extents AdaBoost to the field of SSL by giving pseudo-labels to unlabeled data. It pays more attention to the examples with poor learning effects of the current ensemble learner in each round and continuously improves the robustness using new base learners.

### A.4.2 Deep SSL Algorithms

1. Pseudo Label (Lee, 2013) takes the label with the highest confidence as the pseudo-label and uses cross-entropy obtained from the pseudo-label as the unsupervised loss.

2. Pi-Model (Laine & Aila, 2017) augments the data twice randomly and uses the results of the two augmentations as inputs of the neural network respectively. The inconsistency of the prediction results is used as the unsupervised loss.

3. Mean Teacher (Tarvainen & Valpola, 2017) relies on the idea of knowledge distillation, where the prediction results of the teacher model are used as pseudo-labels to train the student model to ensure the consistency of the prediction results. It uses EMA for the student model's parameters as the teacher model.

4. VAT (Miyato et al., 2018) adds adversarial noise rather than random noise to the data so that the worst performance of the model can be better when the data is affected by noise within a certain range, which corresponds to the zero-sum game in game theory and Min-Max problem in optimization.

5. ICT (Verma et al., 2022) linearly interpolates data and prediction results by Mixup. The unsupervised loss is obtained by the interpolation consistency.

6. UDA (Xie et al., 2020) performs data augmentation on the unlabeled examples and then compares the prediction results before and after the augmentation. The thresholds are used for sample selection for both labeled and unlabeled data respectively.

7. FixMatch (Sohn et al., 2020) uses both strong and weak data augmentation and the inconsistency of prediction results between them is used as the unsupervised loss. A fixed threshold is used for sample selection.

8. FlexMatch (Zhang et al., 2021) uses a dynamic threshold based on FixMatch. It sets a lower confidence threshold for the classes that are more difficult to learn.

9. FreeMatch (Wang et al., 2022b) employs a more precise dynamic threshold, where the threshold setting takes into account both the model's training phase and the disparities between categories. It also incorporates a regularization term to facilitate equitable predictions between categories.

10. SoftMatch (Chen et al., 2023) no longer adheres to the paradigm of filtering examples through confidence threshold, and instead replaces sample selection with sample weighting. The sample weights are utilized to achieve a better balance between the quantity and quality of pseudo-labeled data.

### A.4.3 ROBUST DEEP SSL ALGORITHMS

1. UASD (Chen et al., 2020) ensembles model predictions to produce probability predictions for unlabeled examples, and uses threshold based on confidence to filter out OOD examples.

2. CAFA (Huang et al., 2021) takes into account both the inconsistency in labeling spaces and data distributions. It employs a scoring mechanism to filter out examples from new classes and then utilizes unsupervised domain adaptation to alleviate distribution inconsistency, thus obtaining higher-quality pseudo-labels.

3. MTCF (Yu et al., 2020) leverages the concept of curriculum learning. It uses a joint optimization framework, which updates the network parameters and the OOD score alternately to detect the OOD examples and achieve high performance on the classification simultaneously.

4. Fix-A-Step (Huang et al., 2023) views all OOD unlabeled examples as potentially helpful. It modifies gradient descent updates to prevent optimizing a multi-task SSL loss from hurting labeled-set accuracy.

## A.5 EXPERIMENTS

### A.5.1 DATASETS PREPARATION

Inconsistent Data Distribution

1. Wine, Iris, Letter: To construct datasets with inconsistent distributions, in each class, we calculate the center of all examples and sort these examples according to the distance between them and the center in ascending order. The first $n_c * 0.5$ examples are used as labeled data which can be which can be regarded as being obtained by sampling from $P_0(x, y)$. and the rest of $n_c * 0.5$ examples are used as inconsistent unlabeled data. For each t, the $n_c * 0.5 * (t - \frac{1}{s})$ to $n_c * 0.5 * t$ examples are used as labeled data which can which can be regarded as being obtained by sampling from $P_t(x)$. $\theta(t) = 1$ for every t. 5 examples per class from source domain data are used as labeled data and the rest are used as test data.

2. Image-CLEF: The Image-CLEF dataset consists of 3 domains, which can be combined into 6 source-domain to target-domain pairs. All source domain data can be regarded as being obtained by sampling from $P_{source}(x, y)$ and all target domain data can be regarded as being obtained by sampling from $P_{target}(x, y)$. We set $P_0(x, y) = P_{source}(x, y)$ and $P_t(x, y) = P_{target}(x, y)$ for all $t \neq 0$. From the source-domain data, 100 examples are taken as labeled data. Half of the remaining source-domain examples are used as test data, while the other half is combined with the target-domain data to form an unlabeled dataset. The total number of unlabeled data $n_u$ is $min(0.5 * (n_s - 100), n_t)$ where $n_s$ is the number of examples in the source domain and $n_t$ is the number of examples in the target domain. $\theta(t) = t$ for every t. For inconsistency rate $t$, the unlabeled dataset is combined with $n_u * (1 - t)$ examples for the source domain and $n_u * t$ examples from the target domain.

3. IMDB-Amazon: The IMDB and Amazon datasets can be considered as source and target domains respectively. All source domain data can be regarded as being obtained by sampling from $P_{source}(x,y)$ and all target domain data can be regarded as being obtained by sampling from $P_{target}(x,y)$. We set $P_0(x,y) = P_{source}(x,y)$ and $P_t(x,y) = P_{target}(x,y)$ for all $t \neq 0$. From the source-domain data, 100 examples are taken as labeled data. Half of the remaining source-domain examples are used as test data, while the other half is combined with the target-domain data to form an unlabeled dataset. The total number of unlabeled data $n_u$ is $min(0.5 * (n_s - 100), n_t)$ where $n_s$ is the number of examples in the source domain and $n_t$ is the number of examples in the target domain. $\theta(t) = t$ for every t. For inconsistency rate $t$, the unlabeled dataset is combined with $n_u * (1 - t)$ examples for the source domain and $n_u * t$ examples from the target domain.

## Inconsistent Feature Space

1. Wine, Iris, Letter: $50\%$ of all examples can be used as source domain data, and the rest are used as target domain data. 5 examples per class of source domain data are used as labeled data which can be regarded as being obtained by sampling from $P_0(x,y)$, and the rest are used as test data. For every t, All target domain data randomly dropping $t * d$ features are used as unlabeled data which can be regarded as being obtained by sampling from $P_t(x,y)$. $\theta(t) = 1$ for every t.

2. CIFAR10, CIFAR100: $50\%$ of all examples can be used as source domain data which can be regarded as being obtained by sampling from $P_{source}(x,y)$, and the rest are used as target domain data. 20 examples per class of source domain data are used as labeled data. All target domain data are transformed to grey images by dropping 2 channels which can be regarded as being obtained by sampling from $P_{target}(x,y)$. We set $P_0(x,y) = P_{source}(x,y)$ and $P_t(x,y) = P_{target}(x,y)$ for all $t \neq 0$. $\theta(t) = t$ for every t. For inconsistency rate $t$, the unlabeled dataset is combined with $n_u * (1 - t)$ examples for the source domain and $n_u * t$ examples from the target domain.

3. Agnews: $50\%$ of all examples can be used as source domain data which can be regarded as being obtained by sampling from $P_{source}(x,y)$, and the rest are used as target domain data. 100 examples of source domain are used as labeled data and the rest are used as test data. $50\%$ target domain sentences are used as IID examples and the other $50\%$ target domain sentences that drop $50\%$ tokens are used as OOD examples which can be regarded as being obtained by sampling from $P_{target}(x,y)$. We set $P_0(x,y) = P_{source}(x,y)$ and $P_t(x,y) = P_{target}(x,y)$ for all $t \neq 0$. The number of unlabeled data $n_u$ is set to $min(n_I/(1 - t), n_O/t$ where $n_I$ and $n_D$ are the numbers of IID and OOD examples respectively. The unlabeled dataset is combined with $n_u * (1 - t)$ IID and $n_u * t$ OOD examples. $\theta(t) = t$ for every t.

## Inconsistent Label Space

1. Wine, Iris, Letter: $50\%$ of all examples can be used as source domain data, and the rest are used as target domain data. $(k+1)//2$ classes of source data are saved and the rest examples are dropped which can be regarded as being obtained by sampling from $P_{source}(x,y)$. 5 examples per class of saved source domain data are used as labeled data and the rest are used as test data. The target domain examples with saved classes are used as OOD examples which can be regarded as being obtained by sampling from $P_{target}(x,y)$, and the target examples with dropped classes are used as IID examples. The number of unlabeled data $n_u$ is set to $min(n_I/(1 - t), n_O/t$ where $n_I$ and $n_D$ are the numbers of IID and OOD examples respectively. The unlabeled dataset is combined with $n_u * (1 - t)$ IID and $n_u * t$ OOD examples. We set $P_0(x,y) = P_{source}(x,y)$ and $P_t(x,y) = P_{target}(x,y)$ for all $t \neq 0$. $\theta(t) = t$ for every t.

2. CIFAR10, CIFAR100: $(k+1)/2$ classes of all examples are used as source domain data, and the rest are used as target domain data. 20 examples per class of the source domain are used as labeled data. For inconsistency rate $t$, the unlabeled dataset is combined with $n_t * (1 - t)$ examples for the source domain and $n_t * t$ examples from the target domain where $n_t$ is the number of target domain examples. We set $P_0(x,y) = P_{source}(x,y)$ and $P_t(x,y) = P_{target}(x,y)$ for all $t \neq 0$. $\theta(t) = t$ for every t.

3. Agnews: $(k+1)/2$ classes of all examples are used as source domain data and the rest are used as target domain data. 100 examples of the source domain are used as labeled data. For inconsistency rate $t$, the unlabeled dataset is combined with $n_t * (1-t)$ examples for the source domain and $n_t * t$ examples from the target domain where $n_t$ is the number of target domain examples. We set $P_0(x, y) = P_{source}(x, y)$ and $P_t(x, y) = P_{target}(x, y)$ for all $t \neq 0$. $\theta(t) = t$ for every t.

### A.5.2 Hyper-Parameters of Compared Algorithms

Baselines

1. XGBoost: the parameter of $use\_label\_encoder$ is set to False, the parameter $eval\_metric$ is set to "logloss".
2. FT-Transformers: the number of layers is set to 8, the dimension of tokens is set to 192, and the number of heads is set to 8.
3. ResNet50: the Resnet50 pre-trained on ImageNet from $torchvision.models$ is directly used.
4. Roberta: the pre-trained model "roberta-base" from transformers package is directly used.

Statistical SSL Algorithms

1. SSGMM: the number of iterations is set to 300.
2. TSVM: the parameter $C_l$ is set to 15, the parameter $C_u$ is set to 0.0001, and the method to deal with multi-classification tasks is set to "one vs rest".
3. Label Propagation: the hyperparameters provided by scikit-learn in default are used.
4. Label Spreading: the hyperparameters provided by scikit-learn in default are used.
5. Tri-Training: all the base learners are set to XGBoost classifier consistent with the baseline.
6. Assemble: the number of iterations $T$ is set to 30, and all the base learners are set to XGBoost classifier consistent with the baseline.

Deep SSL Algorithms

1. Pseudo Label: the ratio of unsupervised loss $\lambda_u$ is set to 1.0, and the threshold is set to 0.95.
2. Pi-Model: the ratio of unsupervised loss $\lambda_u$ is set to 1.0, the warmup rate of unsupervised loss $w_u$ is set to 0.4, and the ratio of unsupervised loss $\lambda_u$ is set to $\max(\frac{t}{T \cdot w}, 1.0)$ where $t$ is current iteration and $T$ is the number of iterations.
3. Mean Teacher: the EMA decay is set to 0.999, the warmup rate of unsupervised loss $w_u$ is set to 0.4, and the ratio of unsupervised loss $\lambda_u$ is set to $\max(\frac{t}{T \cdot w}, 1.0)$ where $t$ is current iteration and $T$ is the number of iterations.
4. VAT: the ratio of unsupervised loss $\lambda_u$ is set to 0.3, the ratio of entropy minimization loss $\lambda_{entmin}$ is set to 0.06, the number of iterations for adversarial training $it_{vat}$ is set to 1, the degree of adversarial noise is set to 6.
5. ICT: the ratio of unsupervised loss $\lambda_u$ is set to 100, and the parameter of Beta distribution in Mixup is set to 0.5.
6. UDA: the ratio of unsupervised loss $\lambda_u$ is set to 1.0, the threshold is set to 0.8, and the temperature of softmax is set to 0.4.
7. FixMatch: the ratio of unsupervised loss $\lambda_u$ is set to 1.0, the threshold is set to 0.95, and the temperature of softmax is set to 0.5.
8. FlexMatch: the ratio of unsupervised loss $\lambda_u$ is set to 1.0, the basic threshold is set to 0.95, the temperature of softmax is set to 0.5, and the threshold warmup mechanism is used.
9. FreeMatch: the ratio of unsupervised loss $\lambda_u$ is set to 1.0, the EMA decay is set to 0.999, the threshold is set to 0.95, and the temperature of softmax is set to 0.5.
10. SoftMatch: the ratio of unsupervised loss $\lambda_u$ is set to 1.0, the basic threshold is set to 0.95, the temperature of softmax is set to 0.5, and the distribution alignment mechanism is used.

Robust Deep SSL Algorithms

1. UASD: the ratio of unsupervised loss $\lambda_u$ is set to 1.0, and the threshold is set to 0.95.

2. CAFA: the base SSL algorithm used is Pi Model, the warmup rate of unsupervised loss $w_u$ is set to $\frac{4}{15}$, The perturbation magnitude $\epsilon$ is set to 0.014 and the Beta distribution parameter $\alpha$ is set to 0.75, the warmup rate of adversarial loss $w_a$ is set to $\frac{8}{15}$, the ratio of unsupervised loss $\lambda_u$ is $\exp(-5 \cdot (1 - \min(\frac{t}{T \cdot w_u}, 1.0))^2)$ and the ratio of adversarial loss $\lambda_a$ is $\exp(-5 \cdot (1 - \min(\frac{t}{T \cdot w_a}, 1.0))^2)$ in the $t$-th iteration where $T$ is the number of iterations.

3. MTCF: the ratio of unsupervised loss $\lambda_u$ is set to 75, the temperature $T$ is set to 0.5, and the parameter of Beta distribution in Mixup is set to 0.75.

4. Fix-A-Step: the parameter of Beta distribution in Mixup is set to 0.75, FixMatch is set to the base SSL method, and all the hyperparameters are the same as FixMatch.

Data Augmentation

1. Agnews and IMDB/Amazon: the weak and strong augmentations are synonyms replacements with 1 and 5 words respectively.

2. wine, iris, letter: the weak and strong augmentations are Gaussian noise with 0.1 and 0.2 rates respectively.

3. CIFAR10, CIFAR100, Image-CLEF: the weak augmentation is RandomHorizontalFlip, and the strong augmentation is RandAugment.

Others Hyper-Parameters

1. batch size: the batch size for the IMDB/Amazon dataset is 8, the batch size for the Agnews dataset is 16, the batch size for the Image-CLEF dataset is 32, the batch size for CIFAR10 and CIFAR100 is 64, the batch size for tabular datasets is 64.

2. iteration: the iteration for Image-CLEF dataset is 2000, the iteration for the tabular dataset is set to 1000, the iteration for $ag\_news$ and IMDB/Amazon is set to 5000, the iteration for CIFAR10 and CIFAR100 is 100000.

3. optimizer: the optimizer for all datasets is SGD with a learning rate of 5e-4 and a momentum of 0.9.

4. scheduler: the scheduler for all datasets is CosineWarmup with $num\_cycles$ 7/16.

## A.6 Results under Inconsistent Distribution

The experimental results can be referenced in tables 13 to 17.

## A.7 Results under Inconsistent Feature Space

The experimental results can be referenced in tables 18 to 20.

## A.8 Results under Inconsistent Label Space

The experimental results can be referenced in tables 21 to 23.

Table 13: Evaluation of SSL algorithms using iris dataset under inconsistent data distributions

| Model | AUC | Acc(0) | WA | EVM | VS | RCC |
|---|---|---|---|---|---|---|
| XGBoost | 0.940 | 0.940 | 0.940 | - | - | - |
| TSVM | 0.973 | **0.983** | 0.960 | 0.015 | 0.018 | -0.377 |
| SSGMM | 0.835 | 0.867 | 0.803 | 0.033 | 0.042 | 0.042 |
| Label Propagation | **0.978** | 0.973 | **0.970** | 0.011 | **0.012** | 0.412 |
| Label Spreading | 0.977 | 0.973 | 0.967 | 0.012 | 0.013 | 0.438 |
| Tri-Training | 0.959 | 0.977 | 0.950 | **0.009** | **0.012** | -0.310 |
| Assemble | 0.973 | 0.947 | 0.947 | 0.011 | 0.016 | **0.787** |
| FT-Transformer | 0.980 | 0.980 | 0.980 | - | - | - |
| Pseudo Label | 0.988 | 0.987 | **0.987** | 0.001 | 0.002 | -0.414 |
| Pi-Model | 0.981 | 0.987 | 0.973 | 0.009 | 0.010 | -0.029 |
| Mean Teacher | 0.987 | 0.987 | 0.987 | 0.000 | 0.000 | - |
| VAT | 0.986 | 0.987 | 0.983 | 0.001 | 0.002 | -0.393 |
| ICT | 0.987 | 0.987 | **0.987** | **0.000** | **0.000** | - |
| UDA | 0.986 | 0.987 | 0.983 | 0.002 | 0.002 | -0.414 |
| FixMatch | 0.977 | 0.983 | 0.967 | 0.011 | 0.012 | -0.380 |
| FlexMatch | 0.978 | 0.980 | 0.967 | 0.016 | 0.017 | **0.109** |
| FreeMatch | 0.978 | 0.970 | 0.967 | 0.010 | 0.011 | 0.603 |
| SoftMatch | 0.981 | 0.987 | 0.967 | 0.009 | 0.012 | -0.441 |
| UASD | 0.984 | 0.980 | 0.977 | 0.008 | 0.009 | 0.217 |
| CAFA | **0.989** | 0.990 | **0.987** | 0.002 | 0.002 | -0.414 |
| MTCF | 0.984 | **0.993** | 0.973 | 0.007 | 0.009 | -0.605 |
| Fix-A-Step | 0.987 | 0.987 | 0.983 | 0.003 | 0.003 | -0.338 |

Table 14: Evaluation of SSL algorithms using wine dataset under inconsistent data distributions

| Model | AUC | Acc(0) | WA | EVM | VS | RCC |
|---|---|---|---|---|---|---|
| XGBoost | 0.872 | 0.872 | 0.872 | - | - | - |
| TSVM | 0.944 | 0.955 | 0.925 | **0.012** | 0.018 | -0.68 |
| SSGMM | 0.446 | 0.429 | 0.379 | 0.049 | 0.070 | 0.006 |
| Label Propagation | 0.928 | 0.944 | 0.909 | 0.017 | 0.016 | -0.596 |
| Label Spreading | 0.927 | 0.939 | 0.899 | 0.020 | 0.024 | **0.037** |
| Tri-Training | **0.952** | **0.965** | **0.936** | **0.012** | **0.012** | -0.765 |
| Assemble | 0.941 | 0.939 | 0.912 | 0.014 | 0.017 | -0.604 |
| FT-Transformer | 0.875 | 0.875 | 0.875 | - | - | - |
| Pseudo Label | 0.931 | 0.931 | 0.925 | 0.005 | 0.006 | -0.575 |
| Pi-Model | 0.961 | 0.973 | 0.936 | 0.017 | 0.018 | -0.641 |
| Mean Teacher | 0.920 | 0.920 | 0.920 | 0.000 | 0.000 | - |
| VAT | **0.972** | **0.984** | 0.955 | 0.010 | 0.011 | -0.824 |
| ICT | 0.957 | 0.957 | **0.957** | **0.000** | **0.000** | - |
| UDA | 0.940 | 0.952 | 0.904 | 0.027 | 0.032 | -0.138 |
| FixMatch | 0.958 | 0.971 | 0.928 | 0.027 | 0.031 | -0.362 |
| FlexMatch | 0.948 | 0.955 | 0.915 | 0.027 | 0.029 | -0.326 |
| FreeMatch | 0.903 | 0.907 | 0.853 | 0.031 | 0.039 | 0.204 |
| SoftMatch | 0.960 | 0.976 | 0.944 | 0.015 | 0.019 | -0.134 |
| UASD | 0.912 | 0.912 | 0.912 | 0.000 | 0.000 | - |
| CAFA | 0.876 | 0.864 | 0.848 | 0.028 | 0.036 | **0.249** |
| MTCF | 0.962 | 0.981 | 0.947 | 0.017 | 0.017 | -0.493 |
| Fix-A-Step | 0.963 | 0.979 | 0.939 | 0.013 | 0.013 | -0.893 |

Table 15: Evaluation of deep SSL methods using ImageNet/Catch and ImageNet/Pascal datasets.

| Dataset | Model | AUC | Acc(0) | WA | EVM | VS | RCC |
|---|---|---|---|---|---|---|---|
| ImageNet/Caltech | Supervised | **0.909** | 0.909 | **0.909** | - | - | - |
| | Pseudo Label | 0.907 | 0.908 | 0.907 | 0.001 | 0.001 | -0.621 |
| | Pi-Model | **0.909** | 0.907 | 0.907 | 0.001 | 0.001 | 0.655 |
| | Mean Teacher | 0.903 | 0.904 | 0.900 | 0.003 | 0.003 | 0.169 |
| | VAT | 0.888 | 0.881 | 0.881 | 0.002 | 0.002 | **0.928** |
| | ICT | 0.907 | 0.909 | 0.903 | 0.003 | 0.004 | -0.359 |
| | UDA | 0.896 | 0.904 | 0.891 | 0.006 | 0.007 | -0.512 |
| | FixMatch | 0.902 | 0.905 | 0.887 | 0.005 | 0.007 | -0.726 |
| | FlexMatch | 0.906 | **0.921** | 0.893 | 0.008 | 0.010 | -0.861 |
| | FreeMatch | 0.864 | 0.916 | 0.832 | 0.031 | 0.028 | -0.786 |
| | SoftMatch | 0.904 | 0.908 | 0.891 | 0.007 | 0.007 | -0.805 |
| | UASD | 0.897 | 0.897 | 0.897 | **0.000** | **0.000** | - |
| | CAFA | 0.893 | 0.892 | 0.889 | 0.002 | 0.002 | 0.820 |
| | MTCF | 0.880 | 0.904 | 0.855 | 0.016 | 0.015 | -0.841 |
| | Fix-A-Step | 0.869 | 0.876 | 0.856 | 0.007 | 0.011 | -0.347 |
| ImageNet/Pascal | Supervised | 0.909 | 0.909 | **0.909** | - | - | - |
| | Pseudo Label | 0.906 | 0.908 | 0.901 | 0.004 | 0.004 | -0.375 |
| | Pi-Model | 0.907 | 0.907 | 0.901 | 0.005 | 0.006 | -0.085 |
| | Mean Teacher | 0.904 | 0.909 | 0.900 | 0.003 | 0.003 | -0.813 |
| | VAT | 0.884 | 0.889 | 0.875 | 0.006 | 0.007 | -0.781 |
| | ICT | 0.906 | 0.909 | 0.903 | 0.004 | 0.004 | 0.000 |
| | UDA | 0.894 | 0.904 | 0.883 | 0.013 | 0.017 | 0.173 |
| | FixMatch | 0.901 | 0.905 | 0.889 | 0.013 | 0.017 | -0.528 |
| | FlexMatch | **0.911** | **0.921** | 0.897 | 0.006 | 0.007 | -0.915 |
| | FreeMatch | 0.855 | 0.876 | 0.824 | 0.022 | 0.025 | -0.665 |
| | SoftMatch | 0.888 | 0.908 | 0.839 | 0.014 | 0.018 | -0.846 |
| | UASD | 0.897 | 0.896 | 0.896 | **0.000** | 0.001 | **0.655** |
| | CAFA | 0.890 | 0.897 | 0.887 | 0.003 | 0.003 | -0.768 |
| | MTCF | 0.875 | 0.909 | 0.840 | 0.014 | 0.012 | -0.928 |
| | Fix-A-Step | 0.877 | 0.885 | 0.867 | 0.009 | 0.010 | -0.524 |

Table 16: Evaluation of deep SSL methods using Caltech/ImageNet and Caltech/Pascal datasets.

| Dataset | Model | AUC | Acc(0) | WA | EVM | VS | RCC |
|---|---|---|---|---|---|---|---|
| Caltech/ImageNet | Supervised | 0.945 | 0.945 | 0.945 | - | - | - |
| | Pseudo Label | 0.953 | 0.949 | 0.949 | 0.001 | 0.002 | 0.655 |
| | Pi-Model | 0.950 | 0.945 | 0.945 | 0.001 | 0.002 | 0.655 |
| | Mean Teacher | 0.951 | 0.955 | 0.948 | 0.003 | 0.003 | -0.714 |
| | VAT | 0.953 | 0.959 | 0.945 | 0.005 | 0.005 | -0.895 |
| | ICT | 0.950 | 0.945 | 0.945 | 0.003 | 0.003 | 0.247 |
| | UDA | 0.949 | 0.943 | 0.943 | 0.003 | 0.003 | **0.769** |
| | FixMatch | 0.952 | 0.957 | 0.943 | 0.003 | 0.003 | -0.928 |
| | FlexMatch | 0.960 | **0.965** | 0.953 | 0.007 | 0.007 | -0.498 |
| | FreeMatch | 0.896 | 0.947 | 0.800 | 0.029 | 0.021 | -0.947 |
| | SoftMatch | 0.941 | 0.960 | 0.912 | 0.011 | 0.011 | -0.91 |
| | UASD | **0.962** | 0.963 | **0.961** | **0.000** | **0.001** | -0.655 |
| | CAFA | **0.962** | **0.965** | 0.960 | 0.002 | 0.002 | -0.690 |
| | MTCF | 0.946 | 0.947 | 0.939 | 0.007 | 0.007 | -0.300 |
| | Fix-A-Step | 0.940 | 0.948 | 0.931 | 0.007 | 0.008 | -0.474 |
| Caltech/Pascal | Supervised | 0.945 | 0.945 | 0.945 | - | - | - |
| | Pseudo Label | 0.951 | 0.949 | **0.949** | 0.001 | **0.001** | **0.781** |
| | Pi-Model | 0.950 | 0.945 | 0.945 | 0.001 | 0.002 | 0.655 |
| | Mean Teacher | 0.951 | 0.953 | 0.948 | 0.002 | 0.003 | -0.825 |
| | VAT | 0.944 | 0.956 | 0.936 | 0.006 | 0.006 | -0.678 |
| | ICT | 0.949 | 0.945 | 0.945 | 0.002 | 0.003 | -0.222 |
| | UDA | 0.946 | 0.943 | 0.939 | 0.006 | 0.007 | 0.030 |
| | FixMatch | 0.933 | 0.957 | 0.852 | 0.022 | 0.036 | -0.729 |
| | FlexMatch | 0.952 | **0.965** | 0.937 | 0.007 | 0.007 | -0.920 |
| | FreeMatch | 0.876 | 0.956 | 0.777 | 0.036 | 0.024 | -0.980 |
| | SoftMatch | 0.933 | 0.960 | 0.892 | 0.017 | 0.013 | -0.920 |
| | UASD | **0.962** | 0.963 | **0.961** | **0.000** | **0.001** | -0.655 |
| | CAFA | 0.961 | 0.960 | 0.960 | 0.001 | 0.001 | 0.169 |
| | MTCF | 0.946 | 0.943 | 0.943 | 0.004 | 0.005 | 0.146 |
| | Fix-A-Step | 0.942 | 0.928 | 0.928 | 0.010 | 0.011 | 0.590 |

Table 17: Evaluation of deep SSL methods using Pascal/Caltech and Pascal/ImageNet datasets.

| Dataset | Model | AUC | Acc(0) | WA | EVM | VS | RCC |
|---|---|---|---|---|---|---|---|
| | Supervised | **0.732** | **0.732** | **0.732** | - | - | - |
| | Pseudo Label | 0.726 | 0.724 | 0.724 | 0.001 | 0.002 | 0.804 |
| | Pi-Model | 0.723 | 0.724 | 0.723 | **0.000** | 0.001 | -0.655 |
| | Mean Teacher | 0.727 | 0.728 | 0.721 | 0.007 | 0.008 | -0.398 |
| | VAT | 0.721 | 0.731 | 0.712 | 0.007 | 0.009 | -0.216 |
| | ICT | 0.727 | 0.727 | 0.721 | 0.006 | 0.006 | -0.241 |
| | UDA | 0.731 | 0.728 | 0.725 | 0.003 | 0.003 | **0.916** |
| Pascal/Caltech | FixMatch | 0.698 | 0.713 | 0.611 | 0.027 | 0.043 | -0.628 |
| | FlexMatch | 0.719 | 0.719 | 0.701 | 0.013 | 0.018 | -0.529 |
| | FreeMatch | 0.634 | 0.704 | 0.545 | 0.035 | 0.028 | -0.964 |
| | SoftMatch | 0.694 | 0.691 | 0.676 | 0.011 | 0.013 | -0.486 |
| | UASD | 0.724 | 0.724 | 0.724 | **0.000** | **0.000** | - |
| | CAFA | 0.724 | **0.732** | 0.719 | 0.005 | 0.007 | -0.516 |
| | MTCF | 0.69 | 0.708 | 0.661 | 0.016 | 0.016 | -0.869 |
| | Fix-A-Step | 0.694 | 0.697 | 0.689 | 0.003 | 0.002 | -0.854 |
| | Supervised | **0.732** | **0.732** | **0.732** | - | - | - |
| | Pseudo Label | 0.724 | 0.724 | 0.721 | 0.002 | 0.002 | -0.414 |
| | Pi-Model | 0.724 | 0.724 | 0.723 | 0.001 | 0.001 | 0.497 |
| | Mean Teacher | 0.728 | 0.727 | 0.724 | 0.003 | 0.004 | -0.549 |
| | VAT | 0.715 | 0.721 | 0.707 | 0.007 | 0.007 | -0.665 |
| | ICT | 0.727 | 0.727 | 0.724 | 0.002 | 0.003 | -0.177 |
| | UDA | 0.722 | 0.728 | 0.715 | 0.005 | 0.007 | 0.272 |
| Pascal/ImageNet | FixMatch | 0.717 | 0.713 | 0.713 | 0.004 | 0.004 | 0.425 |
| | FlexMatch | 0.716 | 0.719 | 0.708 | 0.007 | 0.009 | -0.128 |
| | FreeMatch | 0.659 | 0.708 | 0.633 | 0.021 | 0.025 | -0.629 |
| | SoftMatch | 0.694 | 0.691 | 0.688 | 0.003 | 0.004 | **0.763** |
| | UASD | 0.724 | 0.724 | 0.724 | **0.000** | **0.000** | - |
| | CAFA | 0.716 | 0.717 | 0.715 | 0.004 | 0.004 | 0.000 |
| | MTCF | 0.703 | 0.707 | 0.684 | 0.016 | 0.018 | -0.617 |
| | Fix-A-Step | 0.696 | 0.700 | 0.680 | 0.008 | 0.008 | -0.751 |

Table 18: Evaluation of SSL algorithms using iris dataset under inconsistent feature space

| Model | AUC | Acc(0) | WA | EVM | VS | RCC |
|---|---|---|---|---|---|---|
| XGBoost | **0.920** | 0.920 | **0.920** | - | - | - |
| TSVM | 0.887 | **0.937** | 0.850 | 0.023 | 0.021 | -0.929 |
| SSGMM | 0.756 | 0.92 | 0.677 | 0.049 | 0.054 | -0.906 |
| Label Propagation | 0.797 | 0.820 | 0.780 | 0.011 | 0.011 | -0.705 |
| Label Spreading | 0.832 | 0.843 | 0.817 | **0.007** | **0.008** | -0.912 |
| Tri-Training | 0.861 | 0.893 | 0.843 | 0.020 | 0.025 | -0.638 |
| Assemble | 0.917 | 0.907 | 0.907 | 0.015 | 0.018 | **-0.266** |
| FT-Transformer | **0.937** | 0.937 | **0.937** | - | - | - |
| Pseudo Label | 0.922 | 0.920 | 0.920 | 0.001 | 0.002 | -0.293 |
| Pi-Model | 0.913 | 0.910 | 0.910 | 0.003 | 0.004 | -0.293 |
| Mean Teacher | 0.917 | 0.917 | 0.917 | 0.000 | 0.000 | - |
| VAT | 0.918 | 0.923 | 0.907 | 0.007 | 0.010 | -0.345 |
| ICT | 0.923 | 0.923 | 0.923 | **0.000** | **0.000** | - |
| UDA | 0.926 | 0.927 | 0.92 | 0.005 | 0.006 | -0.744 |
| FixMatch | 0.933 | **0.940** | 0.927 | 0.007 | 0.008 | -0.319 |
| FlexMatch | 0.909 | 0.893 | 0.857 | 0.024 | 0.028 | 0.825 |
| FreeMatch | 0.844 | 0.913 | 0.790 | 0.025 | 0.016 | -0.976 |
| SoftMatch | 0.934 | 0.923 | 0.923 | 0.003 | 0.004 | **0.925** |
| UASD | 0.913 | 0.913 | 0.900 | 0.009 | 0.012 | -0.026 |
| CAFA | 0.931 | 0.933 | 0.927 | 0.003 | 0.003 | -0.071 |
| MTCF | 0.900 | 0.860 | 0.860 | 0.014 | 0.017 | 0.845 |
| Fix-A-Step | 0.898 | 0.877 | 0.877 | 0.014 | 0.017 | 0.339 |

Table 19: Evaluation of SSL algorithms using wine dataset under inconsistent feature space

| Model | AUC | Acc(0) | WA | EVM | VS | RCC |
|---|---|---|---|---|---|---|
| XGBoost | 0.846 | 0.846 | 0.846 | - | - | - |
| TSVM | 0.890 | 0.938 | 0.832 | 0.039 | 0.038 | -0.865 |
| SSGMM | 0.500 | 0.722 | 0.359 | 0.096 | 0.110 | -0.606 |
| Label Propagation | 0.800 | 0.930 | 0.703 | 0.075 | 0.087 | **-0.487** |
| Label Spreading | 0.887 | 0.924 | 0.822 | 0.022 | 0.015 | -0.958 |
| Tri-Training | **0.924** | **0.962** | **0.895** | **0.018** | 0.018 | -0.912 |
| Assemble | 0.897 | 0.938 | 0.841 | 0.021 | **0.012** | -0.977 |
| FT-Transformer | 0.938 | 0.938 | 0.938 | - | - | - |
| Pseudo Label | 0.888 | 0.911 | 0.868 | 0.012 | 0.010 | -0.937 |
| Pi-Model | 0.932 | 0.943 | 0.916 | 0.005 | 0.006 | -0.934 |
| Mean Teacher | 0.914 | 0.914 | 0.914 | 0.000 | 0.000 | - |
| VAT | 0.928 | 0.943 | 0.905 | 0.015 | 0.016 | -0.789 |
| ICT | 0.916 | 0.916 | 0.916 | **0.000** | **0.000** | - |
| UDA | 0.924 | 0.935 | 0.881 | 0.03 | 0.035 | **0.326** |
| FixMatch | 0.912 | 0.959 | 0.881 | 0.022 | 0.025 | -0.564 |
| FlexMatch | 0.910 | 0.932 | 0.851 | 0.048 | 0.056 | -0.153 |
| FreeMatch | 0.888 | 0.922 | 0.835 | 0.031 | 0.036 | -0.068 |
| SoftMatch | 0.887 | 0.957 | 0.816 | 0.050 | 0.069 | 0.000 |
| UASD | **0.962** | **0.962** | **0.962** | **0.000** | **0.000** | - |
| CAFA | 0.928 | 0.951 | 0.873 | 0.035 | 0.046 | -0.228 |
| MTCF | 0.905 | 0.951 | 0.881 | 0.015 | 0.012 | -0.913 |
| Fix-A-Step | 0.917 | 0.946 | 0.873 | 0.023 | 0.021 | -0.923 |

Table 20: Evaluation on CIFAR10 and CIFAR100 under inconsistent feature spaces

| Dataset | Method | AUC | Acc(0) | WA | EVM | VS | RCC |
|---|---|---|---|---|---|---|---|
| CIFAR10 | Supervised | 0.473 | 0.473 | 0.473 | - | - | - |
| | Pseudo Label | 0.519 | 0.524 | **0.515** | **0.002** | **0.003** | -0.874 |
| | Pi-Model | 0.500 | 0.511 | 0.485 | 0.007 | 0.007 | -0.882 |
| | Mean Teacher | 0.470 | 0.486 | 0.457 | 0.006 | 0.005 | -0.962 |
| | VAT | 0.501 | 0.550 | 0.466 | 0.020 | 0.018 | -0.880 |
| | ICT | 0.468 | 0.476 | 0.456 | 0.005 | 0.005 | -0.929 |
| | UDA | 0.498 | 0.505 | 0.438 | 0.019 | 0.025 | -0.707 |
| | FixMatch | 0.517 | 0.551 | 0.430 | 0.037 | 0.042 | -0.661 |
| | FlexMatch | 0.552 | 0.607 | 0.431 | 0.041 | 0.039 | -0.921 |
| | FreeMatch | 0.555 | 0.645 | 0.423 | 0.045 | 0.029 | -0.962 |
| | SoftMatch | **0.559** | **0.661** | 0.453 | 0.042 | 0.009 | -0.998 |
| | UASD | 0.481 | 0.486 | 0.479 | 0.003 | **0.003** | -0.625 |
| | CAFA | 0.484 | 0.502 | 0.469 | 0.007 | **0.003** | -0.988 |
| | MTCF | 0.496 | 0.625 | 0.316 | 0.107 | 0.130 | **-0.604** |
| | Fix-A-Step | 0.516 | 0.551 | 0.424 | 0.025 | 0.032 | -0.832 |
| CIFAR100 | Supervised | 0.368 | 0.368 | 0.368 | - | - | - |
| | Pseudo Label | 0.371 | 0.374 | **0.369** | 0.003 | **0.002** | **-0.126** |
| | Pi-Model | 0.367 | 0.369 | 0.365 | 0.003 | **0.002** | -0.147 |
| | Mean Teacher | 0.342 | 0.368 | 0.301 | 0.013 | 0.007 | -0.972 |
| | VAT | 0.364 | 0.384 | 0.342 | 0.008 | 0.005 | -0.985 |
| | ICT | 0.343 | 0.367 | 0.306 | 0.006 | 0.012 | -0.970 |
| | UDA | 0.349 | 0.350 | 0.343 | 0.004 | 0.003 | -0.655 |
| | FixMatch | 0.378 | 0.401 | 0.336 | 0.011 | 0.013 | -0.93 |
| | FlexMatch | 0.376 | 0.397 | 0.331 | 0.019 | 0.016 | -0.799 |
| | FreeMatch | **0.382** | 0.420 | 0.327 | 0.019 | 0.012 | -0.958 |
| | SoftMatch | 0.373 | **0.406** | 0.316 | 0.014 | 0.018 | -0.934 |
| | UASD | 0.368 | 0.364 | 0.364 | **0.002** | 0.003 | 0.256 |
| | CAFA | 0.344 | 0.365 | 0.316 | 0.010 | 0.007 | -0.966 |
| | MTCF | 0.273 | 0.170 | 0.170 | 0.052 | 0.066 | **0.629** |
| | Fix-A-Step | 0.328 | 0.394 | 0.161 | 0.047 | 0.047 | -0.882 |

Table 21: Evaluation of SSL algorithms using iris dataset under inconsistent label space

| Model | AUC | Acc(0) | WA | EVM | VS | RCC |
|---|---|---|---|---|---|---|
| XGBoost | 0.930 | 0.930 | 0.930 | - | - | - |
| TSVM | 0.986 | **1.000** | 0.955 | 0.013 | 0.015 | -0.721 |
| SSGMM | 0.944 | **1.000** | 0.83 | 0.040 | 0.065 | -0.831 |
| Label Propagation | 0.993 | **1.000** | 0.99 | 0.003 | 0.004 | -0.524 |
| Label Spreading | 0.996 | **1.000** | 0.99 | 0.006 | 0.007 | **-0.272** |
| Tri-Training | 0.997 | **1.000** | 0.985 | 0.005 | 0.007 | -0.618 |
| Assemble | **1.000** | **1.000** | **1.000** | **0.000** | **0.000** | - |
| FT-Transformer | **1.000** | **1.000** | **1.000** | | - | - |
| Pseudo Label | 0.994 | **1.000** | 0.990 | 0.003 | 0.004 | **-0.355** |
| Pi-Model | 0.995 | **1.000** | 0.985 | 0.003 | 0.004 | -0.878 |
| Mean Teacher | **1.000** | **1.000** | **1.000** | **0.000** | **0.000** | - |
| VAT | 0.999 | **1.000** | 0.996 | 0.001 | 0.002 | -0.655 |
| ICT | **1.000** | **1.000** | **1.000** | **0.000** | **0.000** | - |
| UDA | 0.996 | **1.000** | 0.990 | 0.002 | 0.002 | -0.924 |
| FixMatch | 0.995 | **1.000** | 0.985 | 0.003 | 0.002 | -0.930 |
| FlexMatch | 0.996 | **1.000** | 0.990 | 0.003 | 0.005 | -0.497 |
| FreeMatch | 0.956 | 0.991 | 0.871 | 0.028 | 0.023 | -0.905 |
| SoftMatch | 0.996 | **1.000** | 0.989 | 0.006 | 0.007 | -0.393 |
| UASD | 0.987 | 0.987 | 0.987 | **0.000** | **0.000** | - |
| CAFA | **1.000** | **1.000** | **1.000** | **0.000** | **0.000** | - |
| MTCF | 0.995 | **1.000** | 0.985 | 0.007 | 0.007 | -0.676 |
| Fix-A-Step | 0.998 | **1.000** | 0.985 | 0.003 | 0.006 | -0.655 |

Table 22: Evaluation of SSL algorithms using wine dataset under inconsistent label space

| Model | AUC | Acc(0) | WA | EVM | VS | RCC |
|---|---|---|---|---|---|---|
| XGBoost | 0.846 | 0.846 | 0.846 | - | - | - |
| TSVM | **0.924** | 0.938 | **0.905** | **0.012** | **0.014** | **-0.721** |
| SSGMM | 0.643 | 0.722 | 0.560 | 0.035 | 0.025 | -0.972 |
| Label Propagation | 0.889 | 0.93 | 0.833 | 0.023 | 0.027 | -0.903 |
| Label Spreading | 0.883 | 0.924 | 0.833 | 0.028 | 0.033 | -0.831 |
| Tri-Training | 0.879 | **0.962** | 0.829 | 0.038 | 0.043 | -0.834 |
| Assemble | 0.904 | 0.938 | 0.862 | 0.021 | 0.018 | -0.929 |
| FT-Transformer | **0.964** | 0.964 | **0.964** | - | - | - |
| Pseudo Label | 0.885 | 0.898 | 0.862 | 0.015 | 0.016 | -0.815 |
| Pi-Model | 0.862 | 0.880 | 0.840 | 0.015 | 0.016 | -0.611 |
| Mean Teacher | 0.920 | 0.920 | 0.920 | **0.000** | **0.000** | - |
| VAT | 0.882 | 0.920 | 0.837 | 0.031 | 0.029 | -0.891 |
| ICT | 0.920 | 0.920 | 0.920 | **0.000** | **0.000** | - |
| UDA | 0.881 | 0.873 | 0.858 | 0.024 | 0.028 | 0.138 |
| FixMatch | 0.879 | 0.913 | 0.833 | 0.016 | 0.009 | -0.978 |
| FlexMatch | 0.871 | 0.927 | 0.833 | 0.031 | 0.036 | -0.648 |
| FreeMatch | 0.831 | 0.877 | 0.790 | 0.026 | 0.038 | -0.601 |
| SoftMatch | 0.943 | **0.976** | 0.920 | 0.013 | 0.011 | -0.911 |
| UASD | 0.923 | 0.923 | 0.923 | **0.000** | **0.000** | - |
| CAFA | 0.880 | 0.815 | 0.815 | 0.042 | 0.043 | **0.779** |
| MTCF | 0.908 | 0.971 | 0.825 | 0.041 | 0.045 | -0.853 |
| Fix-A-Step | 0.947 | 0.960 | 0.927 | 0.007 | 0.004 | -0.976 |

Table 23: Evaluation on CIFAR10 and CIFAR100 under inconsistent label spaces

| Dataset | Method | AUC | Acc(0) | WA | EVM | VS | RCC |
|---|---|---|---|---|---|---|---|
| CIFAR10 | Supervised | 0.643 | 0.643 | 0.643 | - | - | - |
| | Pseudo Label | 0.692 | 0.708 | 0.676 | 0.006 | 0.004 | -0.973 |
| | Pi-Model | 0.672 | 0.703 | 0.654 | 0.01 | 0.009 | -0.937 |
| | Mean Teacher | 0.639 | 0.647 | 0.634 | 0.003 | 0.005 | -0.333 |
| | VAT | 0.697 | 0.734 | 0.661 | 0.015 | 0.011 | -0.974 |
| | ICT | 0.643 | 0.647 | 0.642 | **0.002** | **0.002** | -0.819 |
| | UDA | 0.676 | 0.73 | 0.594 | 0.027 | 0.015 | -0.963 |
| | FixMatch | 0.608 | 0.705 | 0.479 | 0.047 | 0.036 | -0.933 |
| | FlexMatch | 0.731 | 0.806 | 0.614 | 0.038 | 0.02 | -0.965 |
| | FreeMatch | 0.733 | **0.815** | 0.640 | 0.035 | 0.012 | -0.994 |
| | SoftMatch | 0.723 | 0.806 | 0.601 | 0.041 | 0.021 | -0.968 |
| | UASD | 0.644 | 0.641 | 0.641 | **0.002** | **0.002** | **0.404** |
| | CAFA | 0.675 | 0.674 | 0.672 | 0.005 | 0.006 | 0.093 |
| | MTCF | **0.747** | 0.798 | **0.681** | 0.024 | 0.008 | -0.989 |
| | Fix-A-Step | 0.681 | 0.757 | 0.517 | 0.048 | 0.048 | -0.908 |
| CIFAR100 | Supervised | 0.444 | 0.444 | 0.444 | - | - | - |
| | Pseudo Label | 0.453 | 0.459 | **0.448** | 0.002 | 0.002 | -0.973 |
| | Pi-Model | 0.438 | 0.441 | 0.437 | 0.002 | 0.002 | -0.614 |
| | Mean Teacher | 0.442 | 0.437 | 0.437 | **0.001** | 0.002 | 0.817 |
| | VAT | 0.444 | 0.473 | 0.411 | 0.012 | 0.008 | -0.975 |
| | ICT | 0.441 | 0.441 | 0.44 | **0.001** | **0.001** | **0.54** |
| | UDA | 0.416 | 0.434 | 0.381 | 0.011 | 0.01 | -0.906 |
| | FixMatch | 0.392 | 0.496 | 0.25 | 0.049 | 0.024 | -0.975 |
| | FlexMatch | 0.416 | 0.483 | 0.345 | 0.028 | 0.01 | -0.992 |
| | FreeMatch | 0.465 | 0.518 | 0.386 | 0.026 | 0.011 | -0.978 |
| | SoftMatch | **0.473** | **0.521** | 0.407 | 0.023 | 0.008 | -0.981 |
| | UASD | 0.437 | 0.438 | 0.436 | 0.002 | 0.002 | -0.142 |
| | CAFA | 0.441 | 0.442 | 0.44 | **0.001** | **0.001** | -0.055 |
| | MTCF | 0.254 | 0.28 | 0.234 | 0.009 | 0.007 | -0.953 |
| | Fix-A-Step | 0.414 | 0.486 | 0.299 | 0.037 | 0.024 | -0.954 |

