# OpenReview forum: "Realistic Evaluation of Semi-supervised Learning Algorithms in Open Environments"
_ICLR.cc/2024/Conference — ICLR 2024 spotlight_

### Official Review · Reviewer_Et8T · 2023-10-29

**Soundness:** 3 good
**Presentation:** 4 excellent
**Contribution:** 4 excellent
**Rating:** 8
**Confidence:** 5

**Summary:**

This paper addresses the challenge of non-robust performance in semi-supervised learning (SSL) due to inconsistencies between labeled and unlabeled data, especially in open environments with different data sources. The authors introduce the Robustness Analysis Curve (RAC) and related metrics, reshaping the framework for robust SSL to achieve global robustness rather than just local robustness. They evaluate a variety of statistical and deep SSL algorithms across diverse datasets (tabular, image, and text) in three open environment scenarios: inconsistent data distributions, label spaces, and feature spaces, providing a comprehensive benchmark and detailed analysis to enhance the robustness of SSL algorithms in open environments.

**Strengths:**

1. Clarity and Precision in Motivation: The manuscript excels in presenting a clear and well-articulated motivation for the research conducted. The authors emphasize the challenges posed by inconsistencies between labeled and unlabeled data in open environments. They have set a strong foundation for the study, making it easier for readers to understand the significance of the work and the potential impact it could have in advancing the field of robust machine learning.
2. Comprehensive Experimental Evaluation: The manuscript stands out in its thorough and well-structured experimental evaluation. The authors have gone to great lengths to ensure that the performance of various statistical and deep SSL algorithms is rigorously assessed across a wide range of datasets, including tabular, image, and text data. The inclusion of three distinct open environment scenarios (inconsistent data distributions, label spaces, and feature spaces) adds depth to the evaluation, ensuring that the results are comprehensive and reliable. The meticulous design of the experiments and the extensive coverage of different scenarios demonstrate a commitment to empirical rigor, which significantly enhances the credibility and value of the research findings.
3. Solid Theoretical Foundation: The strength of the manuscript is further augmented by its robust theoretical framework. The introduction of the Robustness Analysis Curve (RAC) and the associated metrics offers a novel perspective to approach the problem of robustness in SSL. The detailed theoretical analysis provided in the paper not only supports the empirical findings but also offers deeper insights into the underlying mechanisms that contribute to the robustness of SSL algorithms in open environments. This solid theoretical grounding ensures that the contributions of the manuscript are not just empirical but also provide a conceptual advancement in the understanding of robust SSL.

**Weaknesses:**

1. The manuscript lacks clarity in some of its detail descriptions, such as the explanation of the variable 't', which is not very comprehensible. The author could provide a detailed example to help readers better understand the meaning and function of 't'.
2. The robust semi-supervised learning referred to in the manuscript actually mainly pertains to safe semi-supervised learning. Perhaps utilizing the term "safe semi-supervised learning" would be more suitable and precise in this context.
3. In Figure 2, there are too many methods presented, making it difficult to read. It would be sufficient to showcase a comparison of some representative methods instead.
4. The manuscript introduces numerous evaluation metrics; however, the inherent relationships among these metrics are not very clear. The individual motivations for proposing each of these metrics could be further elucidated.

**Questions:**

The current theoretical results appear to be applicable only when the degrees of inconsistency across the three types of variations are the same. It raises the question of whether the theory would still hold if the degrees of inconsistency were different.

---

> ### Author Response · Authors · 2023-11-20
>
> Dear reviewer:
>
> Thank you for the feedback on our paper. We appreciate the time and effort you have put into reviewing our work. In this rebuttal, we respond to the concerns raised in the reviews.
>
> Regarding the weakness 1. Thank you for your suggestion. In the revised version, we have added illustrative examples for the variable 't'. Please refer to the third line on the second page.
>
> Regarding the weakness 2. Indeed, there is a high degree of overlap between algorithms defined under these two definitions. Safe semi-supervised learning imposes stricter requirements compared to robust semi-supervised learning (i.e., it cannot be weaker than the baseline that only uses labeled data for supervised learning). Given that many of the algorithms we compared cannot achieve a safe level in various environments, we opted for robust semi-supervised learning, which provides a broader description.
>
> Regarding the weakness 3. Thank you for your suggestion. We have addressed our formatting issues in the new version.
>
> Regarding the weakness 4. We discussed the motivation for each metric at the end of the third section of the paper. In the revised version, we have further supplemented the introduction of motivations.
>
> Regarding the question. This is a good question, and we appreciate the insight you provided. In fact, in scenarios where multiple inconsistencies exist, a variable 't' can only be used when the degrees of inconsistency are the same. For situations with varying degrees of inconsistency, we just need to set multiple corresponding variables 't'. In such cases, the theory will still hold by some simple variable substitutions.
>
> Many thanks.

---

> > ### Comment · Reviewer_Et8T · 2023-11-21
> >
> > Thanks to the author's response, which answered my doubts. After reading the author's response and other commenters' comments, I keep my positive review unchanged.

---

### Official Review · Reviewer_ynTZ · 2023-10-30

**Soundness:** 3 good
**Presentation:** 3 good
**Contribution:** 3 good
**Rating:** 8
**Confidence:** 5

**Summary:**

This paper introduces a new robustness evaluation framework for semi-supervised learning, comprising multiple evaluation metrics. It assesses a wide range of semi-supervised learning algorithms including statistical semi-supervised learning algorithms, classical deep semi-supervised learning algorithms, and robust deep semi-supervised learning algorithms in three open environments. This paper also establishes a solid theoretical framework and provides a valuable analysis of the robustness of current semi-supervised learning based on experimental results and the theoretical foundation.

**Strengths:**

1.	The evaluation method employed in this benchmark is reliable, and the adopted metrics are novel and diverse. These reflect the robustness of SSL algorithm performance from different perspectives.
2.	This paper conducted numerous experiments, evaluating a wide set of algorithms which includes commonly used statistical semi-supervised learning algorithms, classical deep semi-supervised learning algorithms, and robust semi-supervised learning algorithms. The experimental results are comprehensive and convincing.
3.	It is good to see that the paper is well-supported with abundant materials, providing extensive theoretical results and detailed descriptions of experiments, rendering it highly reliable.

**Weaknesses:**

It appears that some algorithms were only evaluated on tabular and image datasets, with no evaluation on text datasets.

**Questions:**

Some algorithms have been evaluated on image and tabular datasets but not on text datasets. What could be the reason for this difference?

---

> ### Author Response · Authors · 2023-11-20
>
> Dear reviewer:
>
> Thank you for the feedback on our paper. We appreciate the time and effort you have put into reviewing our work. In this rebuttal, we respond to the concerns raised in the reviews.
>
> Regarding the question and the weaknesses. Certain methods have not been employed in the evaluation of text data, as they were initially designed for image data. For instance, numerous algorithms utilize Mixup operations designed for image data. While these algorithms can be adapted for tabular data to some extent, their applicability to text data lacks generality and rationale. As a result, we did not evaluate them in experiments involving textual data. We will clarify this point in the new version of the paper.
>
> Many thanks.

---

> > ### Comment · Reviewer_ynTZ · 2023-11-21
> >
> > Thanks to the author's response, which addresses my concerns. Having reviewed the author's response along with the comments from other reviewers, I decide to maintain my positive evaluation.

---

### Official Review · Reviewer_dSe9 · 2023-10-31

**Soundness:** 4 excellent
**Presentation:** 3 good
**Contribution:** 4 excellent
**Rating:** 8
**Confidence:** 4

**Summary:**

This paper studies an important aspect of the research field of semi-supervised learning, specifically focusing on semi-supervised learning in open environments. It considers issues such as inconsistent data distributions, label spaces, and feature spaces. The authors build a comprehensive benchmark, including various data types such as tabular data, image data, and text data, and adequate evaluation metrics for this problem. Additionally, the paper provides a solid theoretical analysis to guide future research in this direction.

**Strengths:**

1.	This paper studies an important problem of semi-supervised learning, called robust semi-supervised learning in open environments. This line of research attracts a lot of attention and is becoming more and more important recently. Therefore, building a benchmark for this problem is meaningful and reasonable.
2.	This paper has built a comprehensive benchmark for robust semi-supervised learning in open environments. This benchmark includes various data types, 20 semi-supervised methods, and extensive experimental results and analyses. Therefore, this paper makes a significant contribution to advancing the development of the semi-supervised learning community.
3.	The authors provide both empirical results and theoretical analysis, making this paper technically sound. The provided theoretical framework discusses three different challenges in a unified manner, which are non-trivial and insightful.

**Weaknesses:**

1.	Although the content of this paper is sufficient and informative (up to 46 pages), the main paper contains excessive text and lacks tables and figures to present the experiment results and conclusions.
2.	Section 7 discusses the theoretical results of this paper. It would be more appealing for authors to present some theorems and conclusions in the main paper, rather than presenting only the proof sketch of the theorems.

**Questions:**

1.	This paper proposes various metrics to evaluate the robustness of existing semi-supervised learning methods in different aspects. Is it possible to propose a unified method for ranking existing methods, as there are many experiment results but we cannot know which algorithm is better?
2.	Please refer to some questions raised in the weakness section.

**Details Of Ethics Concerns:**

No ethics concerns.

---

> ### Author Response · Authors · 2023-11-20
>
> Dear reviewer:
>
> Thank you for the feedback on our paper. We appreciate the time and effort you have put into reviewing our work. In this rebuttal, we respond to the concerns raised in the reviews.
>
> Regarding the question 1. It is possible to rank algorithms based on any metric, but the obtained results only represent a specific aspect. For instance, the algorithm with the best average accuracy, the one with the best worst-case accuracy, or the one with the smallest performance variation. In practical applications, it is essential to establish ranking criteria according to the user's requirements for the algorithm. If necessary, comparisons can also involve weighted considerations for different metrics. We summarized the robustness performance of the compared algorithms in Tables 3 and 4. However, these rankings based on metrics only capture a partial aspect of the algorithm's characteristics.
>
> Regarding the weaknesses 1. Thank you very much for your suggestions. We will address formatting issues in the revised version of the paper, addressing the concerns of excessive text in the main body.
>
> Regarding the weaknesses 2. Due to the numerous symbols needed to describe the theorems within the theoretical framework, we currently lack sufficient space in the main text to introduce them. Therefore, we had to place the theorems in the appendix. If there is an opportunity in the future to submit a camera-ready version (with an additional page), we will consider your suggestions.
>
> Many thanks.

---

### Official Review · Reviewer_2gQd · 2023-11-01

**Soundness:** 3 good
**Presentation:** 3 good
**Contribution:** 4 excellent
**Rating:** 8
**Confidence:** 5

**Summary:**

This paper establishes a benchmark for robust semi-supervised learning in open environments, providing reliable evaluation and analysis methods. The proposed evaluation method is used to assess the robustness of current mainstream semi-supervised learning algorithms. The robustness of semi-supervised learning algorithms in open environments is analyzed based on both theoretical considerations and experimental results.

**Strengths:**

S1. Scientifically sound tools and metrics, such as RAC curves and evaluation metrics like AUC, EA, and WA are employed. This has made a positive contribution to the evaluation of existing semi-supervised learning algorithms and the standards for designing future new algorithms.

S2. This research is comprehensive, considering various open environments, data modalities, and types of semi-supervised learning algorithms.

S3. The analysis section is based on a solid theoretical foundation and experimental results, ensuring its reliability.

**Weaknesses:**

The main weakness, as discussed in the limitations section by the authors, lies in the fact that the proposed evaluation framework cannot assess highly complex real-world scenarios with low calculation complexity.

**Questions:**

In practical applications, how can the metrics for measuring real-world inconsistency be correlated with the inconsistency in the evaluation framework?

---

> ### Author Response · Authors · 2023-11-20
>
> Dear reviewer:
>
> Thank you for the feedback on our paper. We appreciate the time and effort you have put into reviewing our work. In this rebuttal, we respond to the concerns raised in the reviews.
>
> Regarding the question. In practical applications, specific metrics may be necessary to quantify inconsistencies. Considering the scenario where all unlabeled samples are inconsistent samples: in the case of inconsistency in data distributions, the degree of inconsistency can be directly measured by numerous metrics in the field of transfer learning such as $\mathcal{H}\Delta\mathcal{H}$ distance; in the case of inconsistency in label or feature spaces, the degree of inconsistency can be directly measured by the proportion of shared classes or features between labeled and unlabeled data compared to the total number of classes or features. Considering the scenario where unlabeled samples containing not only inconsistent samples but also consistent samples: the proportion of inconsistent samples can also be used as the metric to measure the inconsistency. In the end, these inconsistency metrics can be normalized and linked to the variable 't'.
>
> Regarding the weakness. Indeed, evaluation complexity rises with the environment's complexity. In complex environments, we must strike a balance between high precision and low complexity in evaluations.
>
> Many thanks.

---

> > ### Comment · Reviewer_2gQd · 2023-11-21
> >
> > Thank the author for providing an intuitive explanation. After reading the response of the author and comments from other reviewers, I would like to keep my positive rating unchanged.

---

### Meta-Review · Area_Chair_DCnD · 2023-11-27

**Metareview:**

This paper argued that in semi-supervised learning the inconsistency between labeled data and unlabeled data may be from data distributions, label spaces, and/or feature spaces. The paper proposed a set of tools, Robustness Analysis Curve (RAC) and the associated metrics based on RAC, to better evaluate the robustness of existing SSL methods. A lot of experiments were done with many datasets and baselines, showing that existing SSL methods are only partially robust given those three types of inconsistency. Finally, it was considered how to make existing SSL methods more robust in cases with such inconsistency. This paper has received consistently very positive reviews (four "8: accept, good paper" even before the rebuttal), because not only the contributions are solid but also the studied problem is practically meaningful to a lot of researchers in the area of semi-supervised learning. Thus, we should definitely accept it for publication at ICLR 2024.

**Justification For Why Not Higher Score:**

I don't mind if it is bumped up to "Accept (oral)". I just think the topic semi-supervised learning is a bit old and thus may not be very attractive to ICLR participants.

**Justification For Why Not Lower Score:**

This paper has received consistently very positive reviews (four "8: accept, good paper" even before the rebuttal), because not only the contributions are solid but also the studied problem is practically meaningful to a lot of researchers in the area of semi-supervised learning.

---

### Decision · Program_Chairs · 2024-01-16

Accept (spotlight)